# NanoFLUX: Distillation-Driven Compression of Large Text-to-Image Generation Models for Mobile Devices

Ruchika Chavhan [1]   Malcolm Chadwick [1]   Alberto Gil Couto Pimentel Ramos [1]   Luca Morreale [1]
Mehdi Noroozi [1]   Abhinav Mehrotra [1]

## Abstract

While large-scale text-to-image diffusion models continue to improve in visual quality, their increasing scale has widened the gap between state-of-the-art models and on-device solutions. To address this gap, we introduce NanoFLUX, a **2.4B** text-to-image flow-matching model distilled from **17B** FLUX.1-Schnell using a progressive compression pipeline designed to preserve generation quality. Our contributions include: (1) A model compression strategy driven by pruning redundant components in the diffusion transformer, reducing its size from 12B to 2B; (2) A ResNet-based token *downsampling* mechanism that reduces latency by allowing intermediate blocks to operate on lower-resolution tokens while preserving high-resolution processing elsewhere; (3) A novel text encoder distillation approach that leverages visual signals from early layers of the denoiser during sampling. Empirically, NanoFLUX generates $512 \times 512$ images in approximately 2.5 seconds on mobile devices, demonstrating the feasibility of high-quality on-device text-to-image generation.

## 1. Introduction

Large-scale Text-to-Image (T2I) diffusion models have achieved rapid advances in visual quality, primarily driven by the continued scaling of model architectures and training data. Recent examples include Z-Image (Team, 2025), the FLUX family—FLUX.2 (FAL, 2025) and FLUX.1 (BFL, 2025a;b)—which scale up to 17B parameters, as well as Qwen-Image (Wu et al., 2025a), a 20B-parameter model paired with a large 7B-parameter text encoder (Bai et al., 2025). The high visual quality of recent models has driven widespread adoption of text-to-image generation for creative content production across art, design, and media. However, the scale of state-of-the-art systems imposes substantial computational demands, requiring users to either invest in costly GPU infrastructure or rely on cloud-based inference solutions. This imposes substantial barriers for users with limited computational resources. In this work, we seek to reduce these barriers by making high-quality text-to-image models efficiently deployable on mobile devices.

Achieving on-device deployment for large-scale models requires addressing inference latency and model size as a primary constraint. Several approaches to architectural compression in diffusion models focus on exploiting structural redundancy through sparsity (Zhang et al., 2026; Wan et al., 2025; Kalanat et al., 2025), reducing network depth (Kwon et al., 2026; Fang et al., 2024), and quantization (Morreale et al., 2025). In parallel, latency-oriented methods address the quadratic complexity of self-attention. Prior work has explored a range of strategies to mitigate this cost, including linearized attention (Becker et al., 2025; Xie et al., 2025), token merging (You et al., 2025; Lu et al., 2025; Wu et al., 2025b; Bolya & Hoffman, 2023; Ramesh & Zhao, 2024; Zhang et al., 2025b; Guo et al., 2025), and token pruning (Cheng et al., 2025; Zhang et al., 2025a).

In this paper, we introduce NanoFLUX, a **2.4B**-parameter text-to-image diffusion model distilled from a **17B**-parameter FLUX.1-Schnell teacher using a purely distillation-driven compression pipeline. We focus on the two heaviest components of the teacher model, namely the **12B** Diffusion Transformer (DiT) and the **5B** T5-XXL (Chung et al., 2022) text encoder. We structure the entire distillation process as a sequence of progressive stages that incrementally compress the model while preserving generation quality. Our primary contributions are:

1. First, we analyze and exploit multiple sources of architectural redundancy in the diffusion transformer to reduce its size from **12B** to **2B**. Specifically, we compress the model by pruning redundant attention heads, reducing depth via transformer block merging, and precomputing normalization layers where possible.

---

[1]Samsung AI Center, Cambridge. Correspondence to: Ruchika Chavhan <r2.chavan@samsung.com>.

*Proceedings of the $43^{rd}$ International Conference on Machine Learning*, Seoul, South Korea. PMLR 306, 2026. Copyright 2026 by the author(s).

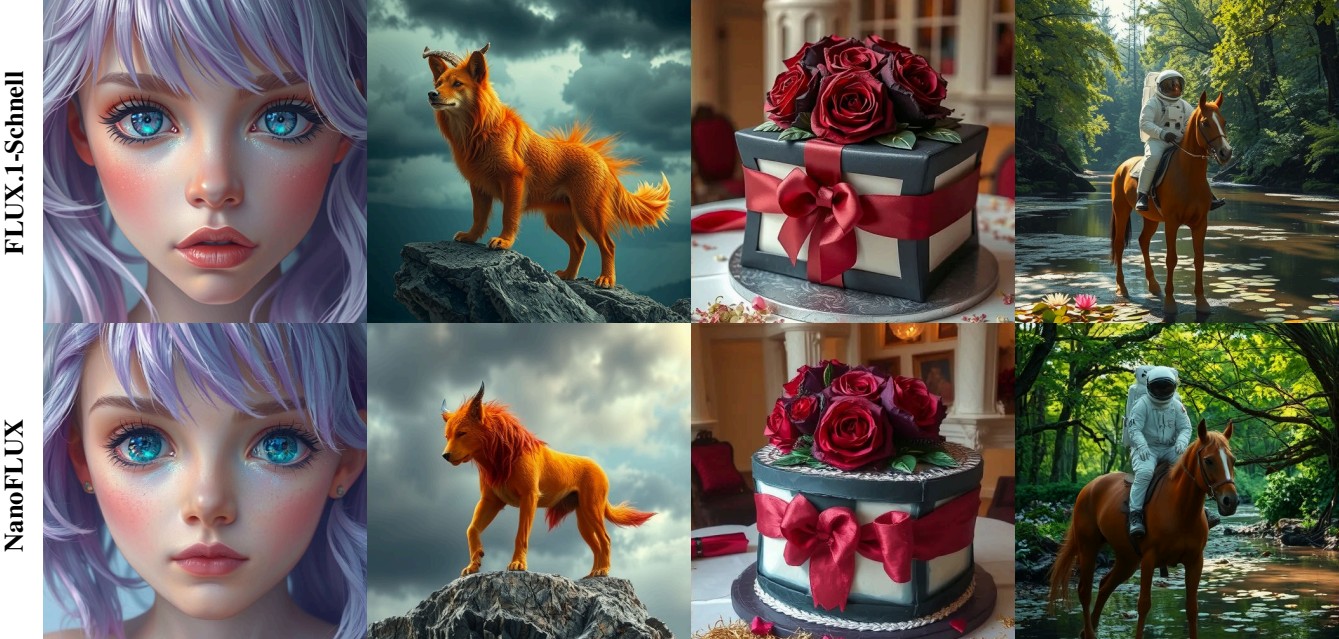

*Figure 1.* Our **NanoFLUX** (**2.4B**), distilled from FLUX.1-Schnell (**17B**) preserves image quality and produces results comparable to the original model, while being $7\times$ smaller.

2. Next, we propose a ResNet-based token downsampling strategy where intermediate layers operate on lower-resolution tokens while preserving high-resolution processing elsewhere, balancing fine-grained detail with coarse representations.

3. Finally, we introduce a novel text encoder distillation method to distill **5B** T5-XXL to a **330M** T5-Large that leverages prompt embeddings from early MMDiT layers, improving semantic alignment and mitigating error accumulation across layers.

Overall, our progressive distillation pipeline reduces the size of the entire model from the **17B** to a **2.4B**-parameter model, while preserving generation quality as shown in Figure 1. We provide an overview of the architecture in Figure 2. We further demonstrate that **NanoFLUX** can generate $512 \times 512$ images in 2.5 seconds on mobile devices, establishing the feasibility of high-quality text-to-image generation under resource-constrained settings. More broadly, our approach provides a practical pathway for scaling down large generative models, broadening their accessibility to the community.

## 2. Related Work

**On-device Image Generation:** Efficient text-to-image generation has been an active area of research in recent years (Shen et al., 2025). Early approaches such as MobileDiffusion (Zhao et al., 2024), FastGAN (Liu et al., 2021), and

SnapFusion (Li et al., 2023) focus on pruning, step reduction, and quantization, but typically operate at much smaller model scales. More recently, SANA-Sprint (Chen et al., 2025) demonstrates impressive speed through step distillation from SANA-1.5 (Xie et al., 2025), but does not explore architectural compression, which is critical for scaling efficiency to large-scale models like FLUX (BFL, 2025a). To the best of our knowledge, NanoFLUX is the first framework to comprehensively study large-scale model compression, offering insights that may generalize to future models.

**Model Compression:** The increasing scale of T2I models has motivated a range of compression techniques that exploit redundant layers, including depth pruning (Kim et al., 2024; Kwon et al., 2026; Castells et al., 2024), low-rank structure (Roy et al., 2026), and dynamic pruning conditioned on timestep (Fang et al., 2024; 2023). In a similar spirit, we introduce a compression framework that first identifies architectural redundancies and subsequently prunes them.

## 3. Preliminaries

**Flow Matching Models** Flow matching models formulate generative modeling as learning a continuous velocity field that transports samples from a simple distribution (like Gaussian noise) to the data distribution. Given an image $x_0 \sim p_{\text{data}}$, where $p_{\text{data}}$ denotes the data distribution, and Gaussian noise $\epsilon \sim \mathcal{N}(0, I)$, a noisy sample at timestep $t \in [0, 1]$ is obtained via linear interpolation as

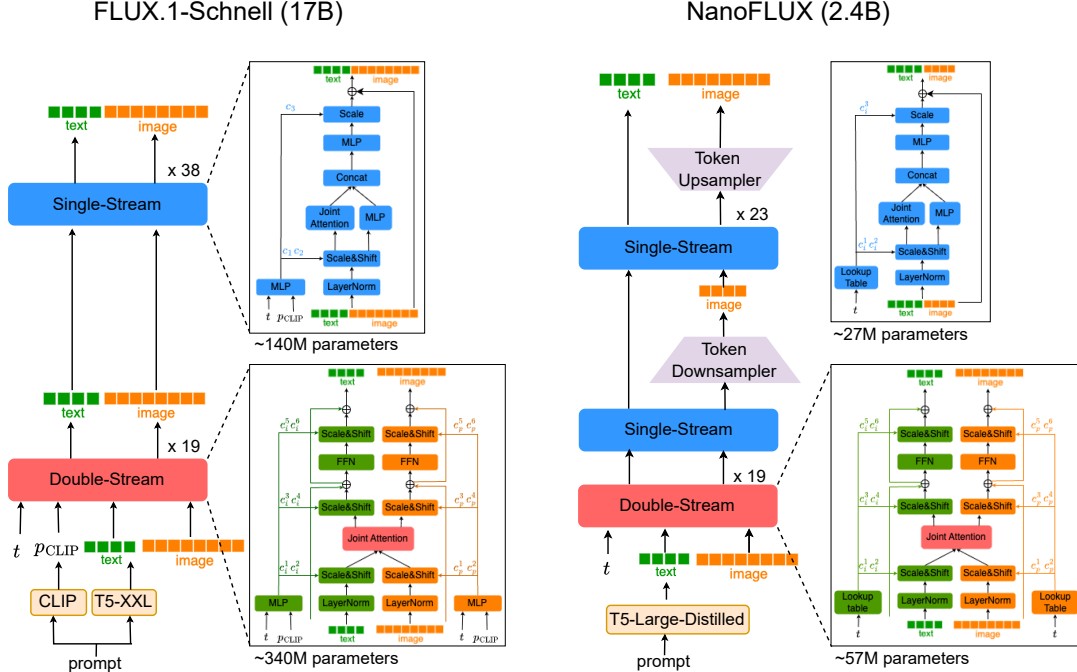

*Figure 2.* Overview of the teacher **17B** FLUX.1-Schnell (left) and **2.4B** NanoFLUX (right).

$x_t = (1 - t)\, x_0 + t\epsilon$. The corresponding target velocity field is then defined as an ordinary differential equation (ODE) $v_t = \frac{dx_t}{dt} = \epsilon - x_0$. For text-to-image generation, a diffusion model $f(\cdot; \theta)$ is trained to predict the velocity at point $x_t$ conditioned on the corresponding prompt $p$ and timestep $t$ by minimizing the flow-matching objective:

$$\mathcal{L}_{\text{FM}} = \mathbb{E}_{x_0, \epsilon, t} \left[ \| f(x_t, t, p; \theta) - (\epsilon - x_0) \|_2^2 \right] \quad (1)$$

At inference time, generation begins by sampling an initial latent $\hat{x}_N \sim \mathcal{N}(0, I)$. Given discrete timesteps $t_0, \cdots t_N \in [0, 1]$, the ODE is solved for $N$ steps as shown in Eq. 2, yielding a final sample $\hat{x}_0$.

$$\hat{x}_{j-1} = \hat{x}_j + \Delta t_j \cdot f(\hat{x}_j, t, p; \theta), \quad \Delta t_j = t_j - t_{j-1} \quad (2)$$

Rather than modeling the data distribution directly in high-dimensional pixel space, images are first mapped to a lower-dimensional latent space using an encoder $\mathcal{E}$, such that a image $y \in \mathbb{R}^{C \times H \times W}$ is encoded as $x_0 = \mathcal{E}(y) \in \mathbb{R}^{c \times h \times w}$. After the denoising process, the resulting latent representation $\hat{x}_0$ is mapped back to the pixel space by a decoder.

### 3.1. Details on FLUX.1-Schnell architecture

FLUX.1 (BFL, 2025b) is a family of rectified flow–based transformer models that achieve state-of-the-art performance in text-to-image generation, combining high visual quality with strong image–text alignment. Black Forest Labs (BFL) first released FLUX.1-Dev, in which classifier-free guidance was distilled from the larger FLUX.1-Pro model. Subsequently, FLUX.1-Schnell was obtained by step distillation from FLUX.1-Dev, reducing inference to four steps. In this work, we adopt FLUX.1-Schnell as our teacher model, allowing us to focus more on architectural-level distillation than step distillation. Below, we describe the core components of FLUX.1-Schnell and outline the overall generation pipeline. An overview of this pipeline can also be found in Figure 2 (left).

**Transformer:** The transformer component of FLUX is based on Multi-Modal Diffusion Transformers (Esser et al., 2024). Unlike UNet-based (Rombach et al., 2021) or Diffusion Transformers (DiTs) (Chen et al., 2024) that incorporate textual guidance in $f(\cdot; \theta)$ through cross-attention, MMDiTs employ *joint* attention over concatenated image and text tokens, enabling stronger and more consistent text–image alignment. MMDiTs are composed of transformer blocks that contain a joint attention module along with separate feed-forward networks (FFNs) and Adaptive Layer Normalization (AdaLN) layers for each modality, as illustrated in Figure 2 (left). As a result, they are referred to as *Double-Stream* blocks in some works. In addition to Double-Stream blocks, the **12B** FLUX.1-Schnell transformer also includes *Single-Stream* (SS) blocks, in which parameters are shared across modalities (See Figure 2).

**Text encoders:** The FLUX pipeline employs two text encoders: T5-XXL (**5B**) and CLIP-Large (**120M**). T5-XXL produces token-level prompt embeddings that are concatenated with image tokens and used in the joint attention mechanism of the diffusion transformer. In contrast, CLIP provides a pooled representation from its final hidden state, which is combined with the timestep embedding and passed to the adaptive normalization layers in the transformer. This dual form of textual guidance contributes to the strong prompt fidelity observed across the FLUX family of models.

The transformer component constitutes the most parameter-heavy part of the architecture, accounting for approximately **12B** parameters, and is therefore the primary focus of our compression efforts. Through a detailed analysis, we identify several architectural redundancies within this transformer. Based on these insights, we design a progressive distillation pipeline that incrementally reduces model size to **2B** while preserving performance. In addition, we distill the **5B**-parameter T5-XXL text encoder into **330M** T5-Large using our proposed text encoder distillation strategy.

**Notations:** Let $H$ denote the number of attention heads and $d_H$ the dimensionality of each head. The model hidden dimension is given by $d = H \cdot d_H$. In practice, the query, key, and value projection matrices in the attention mechanism are of dimension $\mathbb{R}^{d \times d}$. Each FFN consists of a two-layer MLP that projects the $d$-dimensional input to $4d$ dimensions and back to $d$, where the intermediate expansion increases model capacity via width. For FLUX.1 Schnell and all intermediate model variants, we provide the values of $d, H, d_H$ in Table 9 in the Appendix.

## 4. NanoFLUX: Methodology

In this section, we present our progressive distillation strategy. First, we distill the diffusion transformer, from **12B** to **2B** parameters. Then, we distill the text encoder, replacing the T5-XXL **5B** with a **330M**-parameter T5-Large model.

**Training Dataset:** In all experiments, we use prompts from the Ye-PoP dataset (Bassam et al., 2023), a subset of LAION containing approximately 480K images with two captions each. We regenerate high-quality training images using the FLUX.1-Schnell teacher model. To add diversity to the captions, we recaption images using Qwen-Image (Wu et al., 2025a) to generate short, medium, and long descriptions, tags, and a highly detailed caption. During training, we randomly sample one of these prompts for each image. Unless specified, we train the model to generate images of resolution $512 \times 512$. See Section A in the Appendix for details.

**Evaluation metrics:** We evaluate our models on four benchmarks that assess image quality and prompt adherence: One-IG (Chang et al., 2025), DPG (Hu et al., 2024), GenEval (Ghosh et al., 2023), and HPDv3 (Ma et al., 2025). The One-IG benchmark measures performance across five dimensions: alignment, style, diversity, text, and reasoning, where the latter two require text rendering. Since our training data does not include rendered text or captions describing text, we exclude these two categories from the overall score. For HPDv3, we report the Human Preference Score (HPSv3).

### 4.1. Compressing the transformer from 12B → 2B

To compress the DiT model, we rely on a teacher-student framework. In particular, we denote the teacher by $f_T(\cdot; \theta_T)$, parameterized by $\theta_T$, and the student by $f_S(\cdot; \theta_S)$, parameterized by $\theta_S$. In every step of our framework, we minimize knowledge distillation loss:

$$\mathcal{L}_{\text{distill}}(x_t) = \|f_S(x_t, t, p; \theta_S) - f_T(x_t, t, p; \theta_T)\|_2^2, \quad (3)$$

**Optimal inference steps:** We observe a balance between model capacity and the number of inference steps, where lower-capacity student models benefit from more inference steps. Therefore, after training the student model, we perform cross-validation on a held-out validation set to determine the optimal number of inference steps. The selected step counts are reported in Table 9 in the Appendix.

#### 4.1.1. PRUNING ATTENTION LAYERS

In this first stage, we prune the attention layers, leading to a 5B model. Our intuition is that attention heads store redundant information across different layers. Therefore, pruning them will not affect quality significantly.

**Step C1:** 12**B** → 5**B**. We validate our hypothesis through singular value decomposition (SVD) on the head outputs. Specifically, we perform SVD on the $\text{softmax}(QK^\top)V$ term prior to the projection, in each layer. Then, we reconstruct low-rank approximations for every token using the top $r$ singular components and perform inference. As shown in Fig. 4 in the Appendix, for $r = 16$ and $H = 24$ heads, outputs that are highly similar to the original images. Therefore, validating our hypothesis that not all attention heads are independent.

Following our observation, we uniformly reduce the number of attention heads $H$ from 24 to 16 while keeping $d_H$ fixed. This reduces the model dimension to around **5B** parameters. In particular, the student is initialized with the first 16 heads, while other parameters are the same. Empirically, we find that this simple initialization strategy is sufficient to distill a student that closely matches the teacher. For a more in-depth breakdown, see Section B.1 in the Appendix.

To mitigate information loss from head pruning, we introduce an attention-head–level feature loss. Since the conventional feature-level penalty is not applicable, we average the

*Table 1.* **Compressing the DiT 12B → 2B**: We incrementally compress Flux's diffusion transformer from 12B to 1.8B parameters, while relying on T5-XXL as text encoder. First, heads are pruned (**C1**), then the head dimension $d_H$(**C2**). In the last two stages we remove redundant transformer blocks (**C3**), finally we remove AdaLN (**C4**). Image quality is preserved across the distillation stages. See Table 9 in the appendix for details on architecture.

| Model | Steps | One-IG (↑) | DPG (↑) | Geneval (↑) | HPSv3 (↑) |
|---|---|---|---|---|---|
| **FLUX.1-Schnell** | 4 | 43.1 | 84.3 | 66.0 | 11.45 |
| **Step C1: 5B** | 8 | 46.8 | 83.8 | 62.4 | 11.04 |
| **Step C2: 3B** | 10 | 43.2 | 82.3 | 54.1 | 10.74 |
| **Step C3: 2.5B** | 10 | 43.2 | 82.6 | 53.5 | 10.68 |
| **Step C4: 1.8B** | 10 | 43.2 | 82.4 | 53.1 | 10.60 |

attention head's output features over tokens in block:

$$\mathcal{L}_{\text{features}} = \sum_{i=1}^{L} \left\| \frac{1}{H_T} \sum_{h=1}^{H_T} o_T^{l,h}(x_t) - \frac{1}{H_S} \sum_{h=1}^{H_S} o_S^{l,h}(x_t) \right\|,$$
(4)

where $l$ denotes the layer. $H_T$ and $H_S$ denote the number of teacher and student heads and $o$ denotes the output feature. Note that the channels of the student's head $d_H$ remain the same as those of the teacher, while $d$ changes.

The total distillation objective for training the **5B** model is:

$$\mathcal{L}_{5B} = \mathcal{L}_{\text{distill}}(x_t) + \gamma \mathcal{L}_{\text{features}}(x_t)$$
(5)

This distillation objective encourages the remaining heads to collectively preserve the information captured by the original model, enabling effective compression with minimal performance loss. We report the quantitative performance of the **5B** model in Tables 1, observing comparable performance to FLUX.1-Schnell. See Section B.1 and Tables 12, 13, and 14 in the Appendix for more details. Qualitative results are provided in the Appendix in Figure 8.

**Step C2:** 5B → 3B: Motivated by the redundancy highlighted in the previous section, we iterate the SVD analysis on the 5B model. Specifically, we apply SVD to the features of each head, retain the top $r$ singular components, and evaluate the impact on image quality. Figure 5 in the Appendix shows images by varying the value of $r$. The top $r = 96$ components provide the best trade-off between compression and quality. This reduces $d_H$ from 128 to 96, yielding a **3B**-parameter model. Similar to the previous step, we initialize all matrices using a subset of the **5B** model. Tables 1 compare the **3B** model with both the teacher and the **5B** model, showing that it achieves performance comparable to FLUX.1-Schnell while being approximately $1/4$ the size. See Section B.2 and Figure 8 for qualitative comparison.

### 4.1.2. DEPTH PRUNING

Next, we focus on reducing the depth of the **3B** student by identifying and pruning less important transformer blocks.

To this end, we define the importance of a block based on the similarity between its input and output hidden states. Figure 6 in the Appendix visualizes the block-wise input-output cosine similarity, averaged over time steps. High similarity suggests that a block performs only a limited transformation dominated by residual connection from the previous layer, whereas lower similarity indicates a greater contribution towards the global structure. Consistent with prior work (Kwon et al., 2026), we observe that DS blocks demonstrate lower similarity, suggesting that they capture coarse structure, while SS blocks with generally higher similarities tend to refine fine-grained details.

Figure 6 also reveals an interesting pattern: we observe a long contiguous sequence of SS blocks with consistently high similarity scores ($> 0.85$ for image and $> 0.9$ for text tokens), suggesting a large chunk of the model may be functionally redundant. Unlike prior work (Kwon et al., 2026), which prunes such blocks entirely, we *merge* the parameters of redundant blocks by averaging them into a single block. Subsequently, we define our **Step C3: 3B → 2.5B**, where we merge a chain of 15 single-stream blocks (blocks $7 - 22$ which have similarity $> 0.85$ for image tokens) into one, reducing the overall depth of the diffusion transformer from 38 to 24 blocks, resulting in a **2.5B** model. Quantitative results presented in Table 1 show comparable performance to the teacher FLUX.1-Schnell and previous models. We also conduct an ablation study comparing our block merging strategy with standard pruning and iterative pruning approaches, and find that block merging consistently yields better performance. Training details and additional ablations are provided in Section B.3.

### 4.1.3. STATIC LAYER NORMALIZATION

MMDiTs (Esser et al., 2024) employs Adaptive Normalization (AdaLN) (Xu et al., 2019) layers to modulate activations based on conditioning signals: timestep embedding $t \in \mathbb{R}^d$ and pooled CLIP projections denoted by $p_{\text{CLIP}} \in \mathbb{R}^d$. AdaLN in Double-Stream blocks generates six coefficients from $t$ and $p_{\text{CLIP}}$ using a linear projection with $6d^2$ parameters, whereas Single-Stream blocks generate three coefficients using $3d^2$ parameters. This component accounts for a large portion of the model—for example, FLUX.1-Schnell allocates approximately 3.2B parameters to normalization layers. Since the timestep $t$ is shared across samples and textual guidance is already provided by the text encoder, we hypothesize that the additional CLIP-based conditioning contributes limited information and can be removed.

Figure 7 in the Appendix reports the ratio of variance to norm for the coefficients predicted by AdaLN layers, computed over 1,000 samples. We observe that this ratio is consistently small, indicating low variability in the predicted coefficients across samples. Motivated by this observation,

*Table 2.* Applying Static-LN to the **12B**, **5B**, **3B**, and **2.5B** models results in minor performance differences without training. We also present ablations over number of samples used to precompute the coefficients, showing that 2-20 samples are enough, indicating redundancy of a large chunk of model size.

| Model | w Static AdaLN | #Samples | $\Delta$ HPSv3 ($\downarrow$) |
|---|---|---|---|
| 12B | 8.7B | 1000 | 0.01 |
| 5B | 3.6B | 1000 | 0.01 |
| 3B | 2.2B | 1000 | 0.02 |

| Model | w Static AdaLN | #Samples | HPSv3 ($\uparrow$) |
|---|---|---|---|
| 2.5B | X | – | $10.66 \pm 2.58$ |
| 2.5B | 1.8B | 2 | $10.54 \pm 2.67$ |
| 2.5B | 1.8B | 20 | $10.53 \pm 2.67$ |
| 2.5B | 1.8B | 100 | $10.53 \pm 2.68$ |
| 2.5B | 1.8B | 500 | $10.53 \pm 2.69$ |
| 2.5B | 1.8B | 1000 | $10.54 \pm 2.68$ |
| 2.5B | 1.8B (Train) | 1000 | $10.60 \pm 2.62$ |

we replace the dynamic AdaLN coefficients for all 1000 timesteps with average values precomputed over a small calibration set. We refer to our normalization layer as *Static-LN*. This constitutes **Step C4: 2.5B → 1.8B** in our progressive compression pipeline, where we remove 0.7B parameters from the **2.5B** model. As shown in Table 2, we observe very minimal impact on performance of the **2.5B** model *without any training*. Moreover, an ablation study on the number of samples used for precomputation finds that as few as 2 samples are sufficient to achieve comparable performance. Next, fine-tune the **1.8B** model for a few iterations with the precomputed coefficients frozen and observe that quality is largely recovered. Table 1 compares the **1.8B** model with models from previous compression steps. To further validate our approach, we compare the original and Static-LN variants across FLUX.1-Schnell models at **12B**, **5B**, **3B**, and **2.5B** in Table 2, showing only minor differences after precomputing the normalization coefficients.

## 4.2. Progressive Token Downsampling

**Motivation:** The rise of MMDiTs has motivated extensive work on reducing the quadratic cost of self-attention. Prior approaches address this by reducing the number of image tokens through token pruning (Cheng et al., 2025), token merging (Wu et al., 2025b; Bolya & Hoffman, 2023; Smith et al., 2024), attention caching (Yuan et al., 2024), and token downsampling (Tian et al., 2024). Despite differing formulations, these methods typically rely on feature similarity to identify redundant tokens and recover full-resolution outputs via naive token repetition. Moreover, merging and unmerging are often applied at every layer, forcing the entire network to operate on compressed representations, which can be inefficient for large-scale models.

Recent work such as U-DiTs (Tian et al., 2024) draws inspiration from UNet-style multi-scale feature aggregation to introduce latent-space downsampling into transformer architectures, using skip connections to preserve high-resolution information. While effective, these results are demonstrated only for small-scale, class-conditional image generation using DiT architectures without joint attention. As a result, the applicability of such approaches to large-scale text-to-image generation with MMDiTs remains largely unexplored.

In this section, we present Progressive Token Downsampling, a novel approach for token downsampling in text-to-image generation MMDiTs. Our method progressively trains selected transformer blocks to operate on downsampled, low-resolution tokens, while the remaining blocks continue to process high-resolution representations. This hybrid design enables a more favorable trade-off between inference latency and generation quality. We describe the approach in detail below.

### 4.2.1. METHODOLOGY

We consider image hidden states of shape $\mathbb{R}^{T \times d}$ input to a transformer block, where each token corresponds to a localized image region due to the convolutional encoder and pixel shuffle operation (Shi et al., 2016). Reshaping these states into a grid $\mathbb{R}^{\sqrt{T} \times \sqrt{T} \times d}$ aligns neighboring tokens with neighboring spatial regions. With 2D Rotary Position Embeddings (Su et al., 2023), attention explicitly encodes relative 2D offsets, making this spatial structure intrinsic to the model, which we further exploit for token downsampling and upsampling.

**Downsampling and Upsampling layers:** We introduce a ResNet-based downsampling module $D : \mathbb{R}^{\sqrt{T} \times \sqrt{T} \times d} \to \mathbb{R}^{\frac{\sqrt{T}}{2} \times \frac{\sqrt{T}}{2} \times d}$ that reduces the spatial resolution of the latent grid. The resulting features are then reshaped into a sequence of length $\frac{T}{4}$ with dimension $d$, reducing the number of tokens by a factor of 4, thereby accelerating the self-attention. A corresponding ResNet-based upsampling module $U : \mathbb{R}^{\frac{\sqrt{T}}{2} \times \frac{\sqrt{T}}{2} \times d} \to \mathbb{R}^{\sqrt{T} \times \sqrt{T} \times d}$ restores the downsampled tokens to original resolution.

**Where to downsample?:** To preserve high-resolution information in the early DS blocks, we insert the downsampling module $D$ immediately after the first SS block, and apply the upsampling layer $U$ after the final SS block to recover full-resolution representations. In our experiments, we consider the **2.5B** and **1.8B** models from Section 4.1.2 and Section 4.1.3, which comprise 19 DS blocks and 24 SS blocks (see Table 9 in Appendix). As a result, 23 out of the 43 transformer blocks operate on downsampled tokens. Let us denote the blocks between $D$ and $U$ as $\{B_D\}$. We provide an overview in Figure 3 (a) and (b). More architectural details are in Section C in the Appendix.

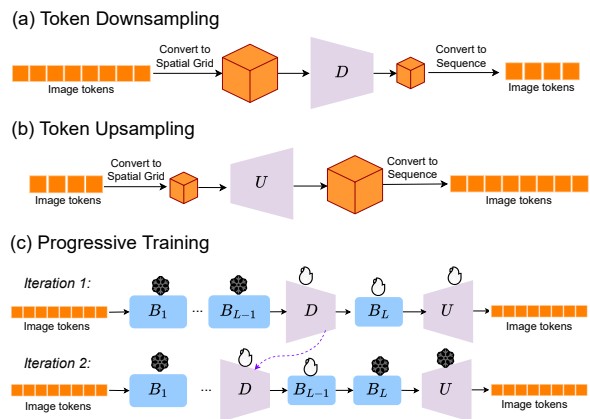

(a) Token Downsampling

(b) Token Upsampling

(c) Progressive Training

*Figure 3.* Overview of Progressive Token Downsampling. (a–b) The ResNet-based downsampler reduces token length and the upsampler restores it. (c) Progressive training enables blocks to operate on downsampled tokens incrementally. Here, $U$ includes the transformer's output projection layers for clarity.

**Hybrid RoPE:** Owing to our hybrid architecture, transformer blocks preceding the downsampling module $D$ operate on high-resolution token grids and therefore use high-frequency 2D RoPE embeddings. In contrast, blocks following $D$ process downsampled tokens and operate on lower-frequency positional information. We denote the high-resolution RoPE embeddings by $R_T$ and the corresponding low-resolution embeddings by $R_{T/4}$.

**When to downsample?:** Intuitively, early diffusion timesteps primarily capture global image structure and can therefore tolerate reduced spatial resolution, whereas later timesteps are responsible for refining fine-grained details and require high-resolution representations. Prior work such as (Tian et al., 2024) does not explicitly account for this distinction, as experiments are conducted at smaller scales where the loss of high-frequency detail is less pronounced.

To address this, we introduce a hybrid strategy in which the transformer blocks in $\{B_D\}$, operate at low resolution during early timesteps and at full resolution during later timesteps. Specifically, we define a timestep threshold $t_{\text{thresh}}$. For $t > t_{\text{thresh}}$, the downsampling module $D$ and upsampling module $U$ are bypassed, and $\{B_D\}$ processes high-resolution tokens with RoPE embeddings $R_T$. For $t \leq t_{\text{thresh}}$, $D$ and $U$ are applied, and $\{B_D\}$ operates on downsampled tokens with low-resolution RoPE embeddings $R_{T/4}$. We next describe our progressive training strategy, which bakes this hybrid resolution scheme into training.

**Progressive Training:** We find that directly inserting the downsampling and upsampling modules, $D$ and $U$, around the block set $\{B_D\}$ leads to unstable training. To address this, we adopt a Progressive Token Downsampling (PTD) strategy, illustrated in Figure 3. We first place $D$ immediately before, and $U$ immediately after the final transformer

*Table 3.* **Progressive Token Downsampling (PTD):** Token downsampling in the Single-Stream (SS) blocks preserves quality while sensibly reducing latency. The downsampler and upsampler add $200M$ parameters to the model.

| Model | PTD | One-IG ($\uparrow$) | DPG ($\uparrow$) | Geneval ($\uparrow$) | HPSv3 ($\uparrow$) |
|---|---|---|---|---|---|
| **Step C3: 2.5B** | X | 43.2 | 82.6 | 53.5 | 10.68 |
| **Step C3: 2.5B** | ✓ | 43.1 | 81.4 | 50.2 | 10.56 |
| **Step C4: 1.8B** | X | 43.2 | 82.4 | 53.1 | 10.60 |
| **Step C4: 1.8B** | ✓ | 43.0 | 81.4 | 49.8 | 10.41 |

block in $\{B_D\}$. For the first iteration, we jointly train $D$, $U$, and the adjacent block while keeping the remainder of the model frozen. We extend the transformer's output projection into a 4-layer MLP, which is also trained during the first training iteration. We then progressively move the downsampling module toward earlier blocks in $B_d$, each time training the newly integrated block together with $D$ while freezing all other components. The upsampling module $U$ and projection layers remain fixed after the initial training stage. This gradual integration stabilizes training and enables effective reduction in token count with minimal performance degradation.

**Results and Ablation Studies:** Table 3 reports the performance of applying PTD to the **2.5B** and **1.8B** models from **Step C3** and **Step C4**, showing minimal quality degradation relative to their full-resolution counterparts. See Figure 10 for qualitative comparison and Section C for training details.

Token merging, pruning, and downsampling are effective in UNet-based diffusion models (Rombach et al., 2021) and show promise in DiT architectures such as PixArt-$\Sigma$ (Yuan et al., 2024), but extending them to MMDiTs with joint text–image attention remains challenging. Related directions, including hybrid linearized attention (Becker et al., 2025) and inference-time feature caching (Zou et al., 2025; Selvaraju et al., 2024), are not directly comparable to our architectural approach. We therefore compare against Linear Attention in SANA-Sprint (Xie et al., 2025) and ablate the timestep threshold, finding $t_{\text{thresh}} = 0.5$ offers the best latency–quality trade-off. (Section C, Table 10 in the Appendix). This results in the first half of the sampling steps using low-resolution tokens, followed by high-resolution tokens in the latter steps.

### 4.3. Text encoder distillation

Now, we discuss our strategy for distillating the **5B** T5-XXL (Chung et al., 2022) into a smaller T5-Large model with **330M** parameters.

**Related Work:** Recent text-to-image models rely on large text encoders such as T5-XXL to better capture complex concepts. To enable on-device deployment, prior work has explored downsizing text encoders through distillation (Wang et al., 2025; Karnewar et al., 2025). These ap-

*Table 4.* Quantitative comparison of NanoFLUX on the OneIG, DPG, Geneval and HPDv3 Benchmarks. Our model NanoFLUX consists of **2B** Diffusion Transformer and a **330M** T5-Large text encoder distilled from **17B** FLUX.1-Schnell pipeline. Qualitative comparisons are shown in Figure 9 in the Appendix.

| Model | DiT | Text | Steps | One-IG ($\uparrow$) | DPG ($\uparrow$) | Geneval ($\uparrow$) | HPSv3 ($\uparrow$) |
|---|---|---|---|---|---|---|---|
| **FLUX.1-Dev** | 12B | T5-XXL (5B) | 20 | 53.5 | 83.8 | 67.0 | 12.01 |
| **FLUX.1-Schnell** | 12B | T5-XXL (5B) | 4 | 43.1 | 84.3 | 66.0 | 11.45 |
| **SANA-1.5** | 1.6B | Gemma (2.6B) | 20 | 42.6 | 84.3 | 65.8 | 10.59 |
| **SANA Sprint** | 1.6B | Gemma (2.6B) | 2 | 43.2 | 63.9 | 73.0 | 10.26 |
| **NanoFLUX** | 2B | T5-Large (330M) | 10 | 42.1 | 75.5 | 49.7 | 10.41 |

proaches distill visual knowledge from the diffusion model into a smaller text encoder, either by distilling for every denoising step (Wang et al., 2025) or by using shallow embeddings before transformer blocks (Karnewar et al., 2025). However, the former may lead to unstable training in high-noise regimes, while the latter does not address error accumulation that may be introduced in further blocks.

In contrast, we focus on intermediate prompt hidden states, which, thanks to joint attention, are directly attended by image tokens and therefore encode visual cues while exhibiting less high-frequency noise than image features. Based on this insight, we propose a text encoder distillation framework that jointly exploits textual and visual signals from prompt embeddings at selected transformer layers.

**Training Method:** Our training procedure consists of two stages, summarized in Algorithm 1 in the Appendix. In the first stage, we attach a two-layer MLP to the student text encoder (T5-Large) to match the teacher's output dimension and train it by minimizing the mean squared error (MSE) between the teacher and student prompt embeddings, denoted by $p_T$ and $p_S$. In the second stage, we perform *block-wise distillation* with a frozen flow-matching diffusion transformer. For each iteration, we sample $x_T \sim \mathcal{N}(0, I)$ and pass both $p_T$ and $p_S$ through the model to collect prompt hidden states at each transformer block. To avoid unstable supervision in high-noise regimes, we sample a cutoff timestep $\hat{t}$ and detach gradients through the student prompt states for $t < \hat{t}$. For later timesteps, we minimize a block-wise MSE between student and teacher-conditioned prompt hidden states. We find that supervising only the first three transformer layers is sufficient. All experiments distill T5-Large using the **2.5B** from **Step C3** diffusion transformer; results in Table 5 show minimal quality degradation. We also observe that the distilled encoder transfers well to the **1.8B** model from **Step C4**.

Next, we compare our method with (Wang et al., 2025), which applies the distillation loss at every timestep and can suffer from unstable gradients. As shown in Table 11 in the Appendix, our block-wise prompt-embedding distillation consistently outperforms this approach.

*Table 5.* **Text Encoder Distillation**: we distill the text encoder to 330M parameters – $6.6\%$ of the original size – while the preserving generation quality.

| Model | T5- | One-IG ($\uparrow$) | DPG ($\uparrow$) | Geneval ($\uparrow$) | HPSv3 ($\uparrow$) |
|---|---|---|---|---|---|
| **C3: 2.5B** | XXL | 43.2 | 82.6 | 53.5 | 10.68 |
| **C3: 2.5B** | Large | 42.1 | 76.3 | 51.8 | 10.45 |
| **C3: 1.8B** | XXL | 43.2 | 82.4 | 53.1 | 10.60 |
| **C4: 1.8B** | Large | 42.1 | 76.2 | 51.7 | 10.45 |

*Table 6.* Latency measurements (in seconds) for the entire denoising process on Samsung S25U.

| | | Compressing the DiT | | | | PTD | |
|---|---|---|---|---|---|---|---|
| Model | **12B** | **5B** | **3B** | **2.5B** | **1.8B** | **2.5B** | **1.8B** |
| Steps | 4 | 8 | 10 | 10 | 10 | 10 | 10 |
| Latency | 14.00 | 4.56 | 3.70 | 2.80 | 2.75 | 2.45 | 2.40 |

### 4.4. End to End Integration of NanoFLUX

Finally, we integrate all our steps. We consider the final **1.8B** model (**Step C4**) from the model compression pipeline (Section 4.1) and train it with Progressive Token Downsampling (Section 4.2), introducing **200M** parameters, resulting in a **2B** DiT. Combined with the distilled T5-Large text encoder (Section 4.3) and the VAE decoder, this produces our final model, **NanoFLUX** (**2.4B**). Figure 2 provides an overview of our architecture. In Table 4, we compare **NanoFLUX** against FLUX.1-Schnell and recent efficient baselines such as SANA-1.5 (Xie et al., 2025) and SANA-Sprint (Chen et al., 2025) (4.2B), showing comparable performance at roughly half the model size.

### 4.5. On-device Latency

To evaluate on-device performance, we deploy NanoFLUX on a Samsung S25U that runs on Qualcomm SM8750-AB Snapdragon 8 Elite Hexagon Tensor Processor. Table 6 reports denoising latency for the **5B**, **3B**, **2.5B**, and **1.8B** models, with and without PTD. The **1.8B** model with PTD achieves a latency of 2.4 seconds. Accounting for the decoder (160 ms) and T5-Large (15 ms), NanoFLUX generates an image in approximately 2.5 seconds end-to-end.

*Table 7.* Quantitative results of training NanoFLUX with $1024 \times 1024$ resolution images.

| Model | One-IG ($\uparrow$) | DPG ($\uparrow$) | Geneval ($\uparrow$) | HPSv3 ($\uparrow$) |
|---|---|---|---|---|
| SANA Sprint | 43.2 | 63.9 | 73.0 | 10.26 |
| FLUX.1-Schnell | 48.3 | 83.2 | 66.5 | 12.14 |
| NanoFLUX | 44.8 | 77.6 | 52.7 | 10.49 |

### 4.6. Training NanoFLUX with higher resolution

Next, we trained NanoFLUX at $1024 \times 1024$ resolution using FLUX.1-Schnell as the teacher model. The model was initialised from the checkpoint obtained during $512 \times 512$ resolution training and subsequently fine-tuned using Progressive Token Downsampling. In this stage, the downsampler $D$ and upsampler $U$ were retained in their respective positions and fine-tuned alongside the intermediate blocks, while the remaining model parameters were kept frozen. T5-XXL was employed as the text encoder throughout this step. As shown in the Table 7, NanoFLUX achieves comparable performance to FLUX.1-Schnell. Furthermore, when compared to the similarly sized SANA-Sprint model, NanoFLUX attains a lower (better) average rank of 1.25 vs 1.75 for SANA-Sprint.

## 5. Conclusions

Through architectural optimizations, progressive token down-sampling strategies and effective text encoder distillation losses, we developed NanoFLUX **2.3B** – a 7x compressed FLUX.1-like high quality model – that generates 512x512 images in 2.5 seconds on mobile devices, narrowing the quality gap between SOTA server models and on-device solutions.

## Impact Statement

This work targets efficient text-to-image generation on edge devices, a key and growing area of research in generative AI. By enabling high-quality image synthesis under constrained compute and memory constraints, our approach broadens access to advanced generative models for users with limited resources or privacy concerns.

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

# A. Dataset Generation

We use prompts from the Ye-PoP dataset to construct a synthetic training set generated with FLUX.1-Schnell. Each Ye-PoP image is associated with two prompts, for which we generate separate images. We then recaption these images using Qwen-Image (7B) (Bai et al., 2025) to increase prompt diversity, producing four types of descriptions: a short summary, a medium-length detailed prompt, object-level tags, and a long caption capturing fine-grained details such as mood, aesthetics, and emotions. We prompt Qwen-Image with the following system template and input image.

> "You are an image captioning assistant. For the given image, generate FOUR different outputs: 1. Short Caption: A concise description (under 15 words) of the image, including main objects, subjects, actions, and location, if applicable. 2. Medium-length Caption: A slightly longer description (15-25 words) focusing on subjects and objects with more details, such as color and position of objects, actions and location, if applicable. 3. Tags: Provide a list of relevant descriptive tags for the image (5-15 tags). Tags should cover objects, subjects, actions, attributes, and context. 4. Descriptive and Aesthetic Caption: Describe the image in detail focussing on the subjects and objects, highlighting the overall style, lighting, color palette and emotion of the image. All the captions should be a single sentence. Do not use very complicated or rare words in any sentence. Output format should be 4 sentences."

# B. Model compression from 12B to 1.8B

In this section, we provide more details on every step of our model compression pipeline defined in Section 4.1.

### B.1. Step C1: 12B → 5B model.

We initialize the **5B** model by pruning redundant attention heads from the **12B** diffusion transformer of FLUX.1-Schnell. As shown in Section 4.1.1 and Figure 4, our singular value decomposition (SVD) analysis of attention head outputs reveals substantial redundancy across heads, motivating a reduction in the number of attention heads.

Specifically, we reduce the number of attention heads $H$ from 24 in the teacher to 16 in the student while keeping the head dimension $d_H$ fixed. This decreases the model dimension from 3072 to 2048, resulting in a model with approximately **5B** parameters. As summarized in Table 9, this change reduces the parameter count of Double-Stream blocks from **340M** to **150M** and Single-Stream blocks from **140M** to **61M**. We implement this head reduction by retaining the first 16 attention heads, corresponding to preserving the top-left $2048 \times 2048$ submatrix of the original $3072 \times 3072$ projection matrices in the attention layers, along with the associated rows in the feedforward layers. Empirically, we find that this straightforward initialization strategy enables the student model to match the teacher's performance after training closely.

**Training Details:** We set $\gamma = 1$ and train this model for 275 epochs with a learning rate of $1e-4$ on a set of 40 H100 GPUs. We observe that the compressed student achieves performance comparable to that of the teacher across evaluation metrics.

### B.2. Step C2: 5B → 3B

To obtain the **3B** model from the **5B** model in the previous step, we extend our analysis to study redundancy across feature dimensions within individual attention heads in the **5B** model. Specifically, we apply SVD to the features of each head, retain the top $r$ singular components, and evaluate the impact on image quality. Figure 5 shows images by varying the value of $r$. We find that keeping the top $r = 96$ components provides the best trade-off between compression and quality, and accordingly reduces $d_H$ from 128 to 96, yielding a **3B**-parameter model. As shown in Table 9, this reduces the size of DS blocks from **150M** to **85M** and SS blocks are reduced from **61M** to **35M**. Similar to **Step C1**, we initialize all matrices by retaining the top-left $1536 \times 1536$ submatrix of the $2048 \times 2048$ projection matrices in the 5B model.

**Training Details:** We train this model for 50 epochs using the **5B** student in **Step C1** as the teacher with a learning rate of $1 \times 10^{-4}$, followed by an additional 100 epochs of distillation from FLUX.1-Schnell **12B** using a cosine annealing learning rate scheduler. To train this model, we rely only on the knowledge distillation loss $\mathcal{L}_{\text{distill}}(x_t)$. This model required around 15 days to train completely. We observe that the compressed student achieves performance comparable to both teachers across evaluation metrics.

| Strategy | Prune | Prune | IterPrune | IterPrune | Merging | Merging |
|---|---|---|---|---|---|---|
| Block Type | DS | SS | DS | SS | DS | SS |
| Block IDs | 7-12 | 7-12 | 7-12 | 7-12 | 7-12 | 7-12 |
| HPSv3 | $9.41 \pm 2.45$ | $9.66 \pm 2.62$ | $9.81 \pm 2.45$ | $10.12 \pm 2.87$ | $9.71 \pm 2.91$ | $10.66 \pm 2.62$ |

*Table 8.* Ablation over pruning and merging transformer blocks such that total model size is **2.5B** for all experiments. Block ID denotes blocks that have been modified. We observe that block merging consistently outperforms block pruning. Moreover, we observe that pruning Double-Stream blocks worsens the quality of the model.

*Table 9.* Architecture details of progressively compressed models in Section 4.1

| Model | # Total Params | Double-Stream | | Single-Stream | | $H$ | $d_H$ | $d$ | Steps |
|---|---|---|---|---|---|---|---|---|---|
| | | #Blocks | #Params | #Blocks | #Params | | | | |
| **FLUX.1-Schnell** | 12B | 19 | 340M | 38 | 140M | 24 | 128 | 3072 | 4 |
| **Step C1: Reducing $H$** | 5B | 19 | 150M | 38 | 61M | 16 | 128 | 2048 | 8 |
| **Step C2: Reducing $d_H$** | 3B | 19 | 85M | 38 | 35M | 16 | 96 | 1536 | 10 |
| **Step C3: Depth pruning** | 2.5B | 19 | 85M | 24 | 35M | 16 | 96 | 1536 | 10 |
| **Step C4: Static-LN** | 2B | 19 | 57M | 24 | 27M | 16 | 96 | 1536 | 10 |

## B.3. Step 3: 3B $\rightarrow$ 2.5B

In this stage of our progressive distillation pipeline, we further compress the **3B** model from **Step C**2 by identifying and eliminating redundancy among transformer blocks, as shown in Figure 6. Rather than removing redundant blocks, we merge their parameters into a single transformer block and apply distillation to preserve functionality during compression.

**Training Details:** We train this model using the knowledge distillation loss $\mathcal{L}_{\text{distill}}$ with the **3B** model as the teacher for 50 epochs with a learning rate of $1 \times 10^{-4}$, followed by further distillation from the 12B FLUX.1-Schnell model using a learning rate of $2 \times 10^{-5}$ for 25 epochs. This model required around 2 days to train completely.

**Ablation Study:** We compare our block merging strategy against two baseline approaches: (1) Pruning (Prune) the entire sequence of redundant blocks at once, and (2) Iterative Pruning (IterPrune), one at a time. Table 8 shows that merging the parameters of redundant blocks consistently yields the best results. We further observe that both pruning and merging operations are more effective when applied to Single-Stream DiT blocks, while modifying MMDiT blocks to obtain a **2.5B** model leads to noticeable performance degradation. Table 4 also reports quantitative comparison of the **3B** model with models from previous compression stages, demonstrating that our block merging strategy effectively reduces model depth while preserving generation quality.

## C. Details on Progressive Token Downsampling

In this section, we provide details on our Progressive Token Downsampling method proposed in Section 4.2. We start by mentioning training details, followed by discussing ablations.

**Architecture Details:** We employ a ResNet-based downsampling and upsampling module that changes the spatial resolution by a factor of two. The downsampler consists of a ResNet block followed by a $3 \times 3$ convolution with stride 2, halving the spatial resolution and reducing the token sequence length by a factor of four. The upsampler applies a ResNet block, bilinear interpolation, and a convolution layer, followed by a second ResNet block. Both modules incorporate timestep-dependent

*Table 10.* We also perform an ablation over $t_{\text{thresh}}$ to select optimal threshold to balance latency and quality. $t_{\text{thresh}} = 0$ implies no compression is applied. We observe an optimal point around $t_{\text{thresh}} = 0.5$, which results in good performance and better latency than $t_{\text{thresh}} = 0.2$, where the model only performs downsampling on the first 20% time steps.

| | $t_{\text{thresh}} = 0$ | $t_{\text{thresh}} = 0.2$ | $t_{\text{thresh}} = 0.5$ | $t_{\text{thresh}} = 1.0$ | LA |
|---|---|---|---|---|---|
| HPSv3 | | 10.66 | 10.61 | 10.56 | 7.89 |

*Table 11.* We compare our block-wise text encoder distillation for replacing T5-XXL (5B) with T5-Large (330M) against (Wang et al., 2025), where $\mathcal{L}_{\text{init}}$ and $\mathcal{L}_{\text{block}}$ denote Stage 1 and Stage 2 losses in Algorithm 1.Leveraging visual cues from early transformer blocks consistently outperforms output-only distillation.

| Model | 2.5B | | | 1.8B | | |
|---|---|---|---|---|---|---|
| Loss | (Wang et al., 2025) | $\mathcal{L}_{\text{warmup}}$ | $\mathcal{L}_{\text{block}}$ | (Wang et al., 2025) | $\mathcal{L}_{\text{init}}$ | $\mathcal{L}_{\text{block}}$ |
| HPSv3 | 10.37 | 10.42 | 10.45 | 10.21 | 10.4 | 10.45 |

scale and shift conditioning. For both the downsampler and upsampler, we set the number of channels equal to the feature dimension of the **2.5B** model, namely $d = 1536$.

**Training Details:** We adopt a progressive training strategy (Figure 3) in which the downsampler $D$ is initially placed immediately before, and the upsampler $U$ immediately after, the final transformer block in $B_D$. In the first stage, we jointly train $D, U$, and the adjacent transformer block while freezing the rest of the model. During this stage, we also extend the transformer's output projection to a four-layer MLP and train it jointly. We then progressively move $D$ toward earlier blocks in $B_D$, training each newly integrated block together with $D$ while keeping all other components fixed. The upsampler $U$ and projection layers remain frozen after the initial stage. Each progressive stage minimizes the distillation loss $\mathcal{L}_{\text{distill}}$ for 80 epochs with a learning rate of $1 \times 10^{-4}$, followed by an additional 20 epochs of end-to-end fine-tuning once all blocks in $B_D$ have been trained with downsampled tokens. Each iteration required 10 hours to train with 48 H100s.

**Ablation over timestep threshold:** We ablate the timestep threshold $t_{\text{thresh}}$ to identify the best trade-off between latency and quality. Setting $t_{\text{thresh}} = 0$ disables downsampling, while $t_{\text{thresh}} = 1$ applies downsampling at all timesteps. Smaller values favor quality with limited speedup, whereas larger values increase latency gains at the cost of quality. As shown in Table 10, $t_{\text{thresh}} = 0.5$ provides the best balance, with the first half of diffusion steps using downsampled tokens and the latter half operating at full resolution to recover fine details.

**Comparison with Linear Attention:** We compare our approach with linear attention, as used in SANA-Sprint (Chen et al., 2025), which linearizes self-attention to achieve significant speedups. Table 10 reports the corresponding HPS scores. While linear attention reduces latency, we find that in FLUX it leads to a substantial drop in image quality. Although recent work has explored hybrid linear attention for models such as SD3.5 (Becker et al., 2025), we observe a persistent gap between linear and softmax attention in modeling long-range dependencies and fine local details. Consequently, we retain softmax attention and focus on token downsampling for efficiency gains.

## D. Text-encoder distillation

**Training Algorithm:** Please refer to Algorithm 1 for an overview of our text encoder distillation strategy. Our training strategy consists of two stages. In Stage 1, we perform warm up training by minimising the MSE loss between $p_T$ and $p_S$. We denote this loss by $L_{\text{warmup}}$. In Stage 2, we first sample a random latent $x_T$ and random cutoff timestep between 0 and $T$, where $T$ is the number of denoising time steps. For $t < \hat{t}$, we cutoff gradients to avoid supervision from high noise signals. For $t > \hat{t}$, we collect teacher-condition and student-conditioned prompt hidden states from transformer blocks and minimise the MSE between them. We denote this loss by $\mathcal{L}_{\text{block}}$

We find that setting $\alpha_1 = \alpha_2 = \alpha_3 = 0.1$ and all remaining coefficients to zero, i.e., supervising only the first few layers, is sufficient, and substantially reduces training time compared to (Wang et al., 2025). We hypothesize that this is due to the presence of a super-weight (Yu et al., 2025) in the third Double-Stream block, which sharply increases the norm of the prompt hidden states. This behavior is also evident in Figure 6, where the input–output similarity for prompt states in Double-Stream layers jumps from approximately 0.3 in the second layer to nearly 1.0 in the third. Our intuition is that matching the prompt norm at this stage is critical for limiting error accumulation in subsequent layers, explaining the effectiveness of our approach.

**Comparison with Baselines:** We compare our text encoder distillation approach with (Wang et al., 2025), which applies output-level distillation at every timestep. As shown in Table 11, our method consistently outperforms this baseline.

# E. Detailed Results

We evaluate all models on four benchmarks: OneIG (Chang et al., 2025), DPG (Hu et al., 2024), GenEval (Ghosh et al., 2023), and HPDv3 (Wu et al., 2023). For each stage of our progressive distillation pipeline, we report the metrics across subcategories of the remaining metrics. Detailed results are provided in Tables 12, 14, and 13.

---

**Algorithm 1** Our proposed text-encoder distillation algorithm.

---

**Require:** Flow-matching model $f$ with blocks $\{B_i\}_{i=1}^L$, Teacher text encoder $\text{T5}_{\text{XXL}}$, Student encoder $\text{T5}_L$ with parameters $\theta_S$, Coefficients $\alpha_i$, Prompt $p$

1: **Stage 1: Student Warm-up**
2: $p_T \leftarrow \text{T5}_{\text{XXL}}(p)$
3: $p_S \leftarrow \text{T5}_L(p; \theta_S)$
4: Minimize training loss $\mathcal{L}_{\text{init}}(p_S, p_T)$

5: **Stage 2: Block-wise Distillation + Denoising Rollout**
6: Sample random cutoff timestep $\hat{t} \sim \mathcal{U}(\{1, \ldots, T\})$
7: Sample latents $x_T \sim \mathcal{N}(0, I)$
8: $\mathcal{L}_{\text{block}} \leftarrow 0$
9: **for** $t = T \rightarrow 1$ **do**
10:     **Student forward (velocity flow + internal states):**
11:     $\hat{v}_t^S, \{h_t^i(p_S)\}_{i=1}^L \leftarrow f(x_t, p_S, t)$
12:     **Teacher forward (internal states only):**
13:     $\_, \{h_t^i(p_T)\}_{i=1}^L \leftarrow f(x_t, p_T, t)$
14:     **if** $t < \hat{t}$ **then**
15:         **for** $i = 1$ **to** $L$ **do**
16:             $h_t^i(p_S) \leftarrow \text{stopgrad}(h_t^i(p_S))$
17:         **end for**
18:     **else**
19:         **for** $i = 1$ **to** $L$ **do**
20:             $\mathcal{L}_{\text{block}} \mathrel{+}= \alpha_i \cdot \left\| h_t^i(p_S) - h_t^i(p_T) \right\|_2^2$
21:         **end for**
22:     **end if**
23:     **Euler Scheduler update:**
24:     $x_{t-1} \leftarrow x_t + \Delta\tau_t\, \hat{v}_t^S$
25: **end for**
26: Update parameters $\theta_S$ using $\mathcal{L}_{\text{block}}$

---

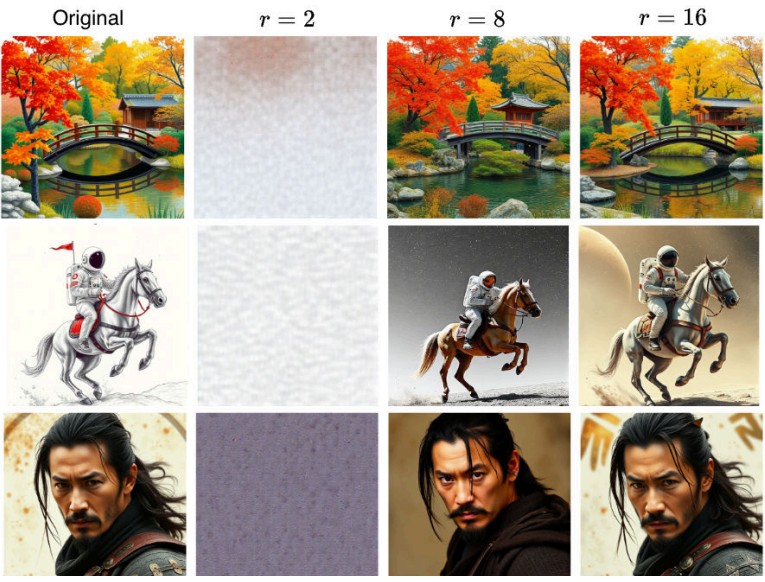

*Figure 4.* Attention head redundancy analysis. Low-rank reconstructions of per-token attention outputs softmax$(QK^\top)V$ using the top $r$ singular components show that $r = 16$ (out of $H = 24$ heads) preserves image quality in FLUX.1-Schnell (12B), indicating substantial redundancy across attention heads.

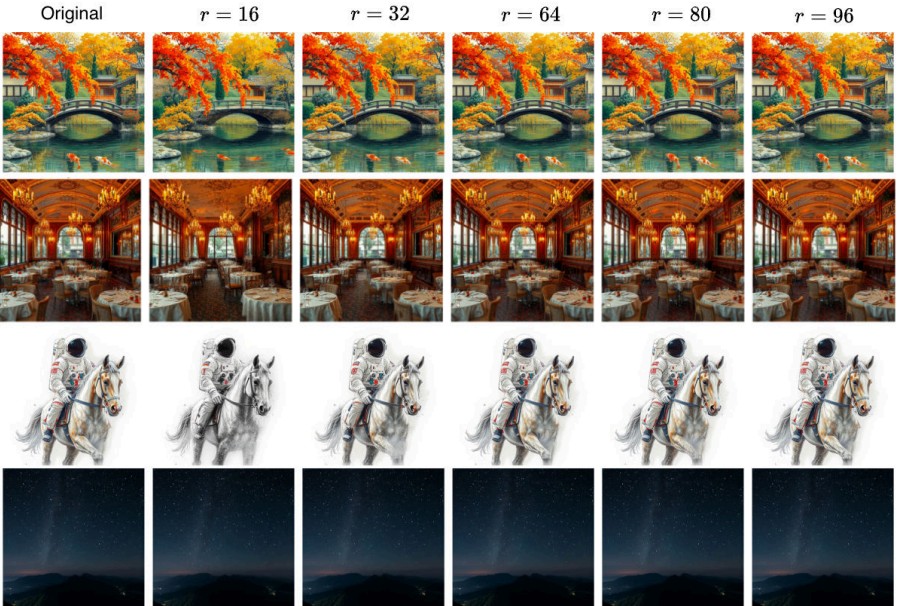

*Figure 5.* Analysing redundancy in features. Low-rank reconstructions of per-head attention outputs $(QK^\top)V$ using the top $r$ singular components show that $r = 96$ preserves image quality in our **5B**, indicating substantial redundancy across features.

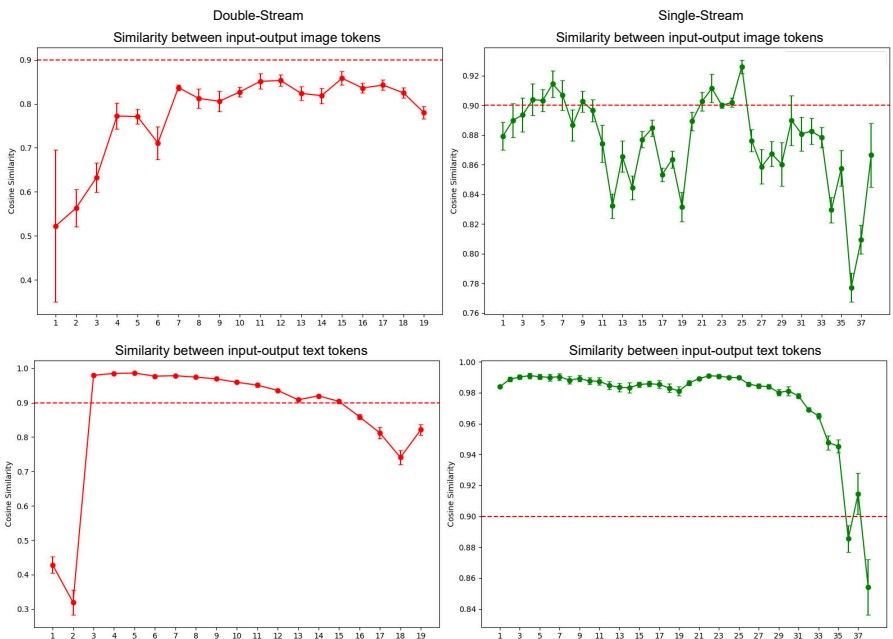

*Figure 6.* Cosine similarity between input and output of transformer blocks in the **3B** model. We observe a sequence of Single-Stream blocks (7-23) that exhibit high similarity, indicating potential redundancy.

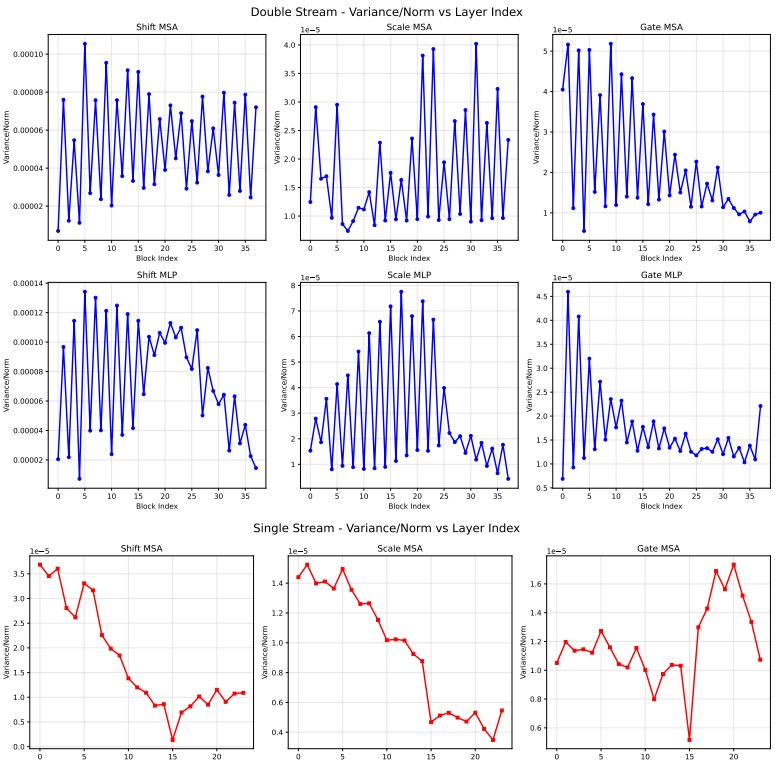

*Figure 7.* Ratio of variance to norm for AdaLN coefficients in DS and SS blocks, calculated over 1000 samples. We observe very low variance over our calibration set which indicates that these can be precomputed.

*Table 12.* Quantitative comparison of all steps in our progressive distillation pipeline on the One-IG Benchmark (Chang et al., 2025) and HPSv3 score on the HPD Dataset (Ma et al., 2025).

| **Compressing the DiT 12B $\rightarrow$ 2B** (Text Encoder: T5-XXL) | | | | | |
|---|---|---|---|---|---|
| Model | Alignment | Style | Diversity | Overall (↑) | HPSv3 (↑) |
| **Step C1: 5B** | 49.6 | 34.5 | 43.6 | 46.8 | $11.04 \pm 2.44$ |
| **Step C2: 3B** | 48.6 | 33.8 | 47.3 | 43.2 | $10.74 \pm 2.59$ |
| **Step C3: 2.5B** | 48.7 | 33.6 | 47.2 | 43.2 | $10.68 \pm 2.58$ |
| **Step C4: 1.8B** | 48.5 | 33.6 | 47.5 | 43.2 | $10.60 \pm 2.62$ |

| **Progressive Token Downsampling (PTD)** | | | | | |
|---|---|---|---|---|---|
| Model | Alignment | Style | Diversity | Overall | HPSv3 |
| **Step C3: 2.5B** | 48.7 | 33.6 | 46.9 | 43.1 | $10.56 \pm 2.87$ |
| **Step C4: 1.8B** | 48.4 | 33.5 | 47.1 | 43.0 | $10.41 \pm 2.87$ |

| **Text Encoder Distillation** | | | | | |
|---|---|---|---|---|---|
| Model | Alignment | Style | Diversity | Overall | HPSv3 |
| **Step C3: 2.5B** | 46.1 | 32.5 | 47.6 | 42.1 | $10.45 \pm 2.73$ |
| **Step C4: 1.8B** | 45.9 | 32.6 | 47.9 | 42.1 | $10.45 \pm 2.73$ |

*Table 13.* Quantitative comparison of all steps in our progressive distillation pipeline on the DPG Benchmark (Ma et al., 2025).

| **Compressing the DiT 12B $\rightarrow$ 2B** (Text Encoder: T5-XXL) | | | | | | |
|---|---|---|---|---|---|---|
| Model | Global | Entity | Attribute | Relation | Other | Overall (↑) |
| **Step C1: 5B** | 83.4 | 84.4 | 83.6 | 83.3 | 83.8 | 83.8 |
| **Step C2: 3B** | 81.7 | 82.4 | 82.0 | 82.3 | 82.3 | 82.2 |
| **Step C3: 2.5B** | 81.7 | 82.7 | 82.5 | 82.6 | 82.3 | 82.3 |
| **Step C4: 1.8B** | 81.1 | 82.0 | 82.4 | 82.4 | 82.0 | 82.0 |

| **Progressive Token Downsampling (PTD)** | | | | | | |
|---|---|---|---|---|---|---|
| Model | Global | Entity | Attribute | Relation | Other | Overall (↑) |
| **Step C3: 2.5B** | 81.0 | 81.0 | 81.4 | 81.4 | 81.5 | 81.4 |
| **Step C4: 1.8B** | 80.9 | 81.3 | 81.4 | 81.8 | 81.6 | 81.4 |

| **Text Encoder Distillation** | | | | | | |
|---|---|---|---|---|---|---|
| Model | Global | Entity | Attribute | Relation | Other | Overall (↑) |
| **Step C3: 2.5B** | 75.8 | 77.4 | 75.3 | 76.8 | 76.3 | 76.3 |
| **Step C4: 1.8B** | 75.7 | 77.3 | 75.3 | 76.3 | 76.1 | 76.2 |

*Table 14.* Quantitative comparison of all steps in our progressive distillation pipeline on the Geneval Benchmark (Ghosh et al., 2023).

| **Compressing the DiT 12B → 2B** (Text Encoder: T5-XXL) | | | | | | | |
|---|---|---|---|---|---|---|---|
| Model | Single Object | Two Objects | Count | Color | Position | Attribute | Overall (↑) |
| **Step C1: 5B** | 97.8 | 79.8 | 45.0 | 77.7 | 25.0 | 48.0 | 62.4 |
| **Step C2: 3B** | 87.5 | 68.7 | 43.8 | 69.2 | 23.0 | 32.0 | 54.1 |
| **Step C3: 2.5B** | 87.5 | 66.7 | 43.8 | 70.2 | 20.0 | 33.0 | 53.5 |
| **Step C4: 1.8B** | 88.8 | 62.6 | 40.0 | 67.0 | 20.0 | 40.0 | 53.1 |
| **Progressive Token Downsampling (PTD)** | | | | | | | |
| Model | Single Object | Two Objects | Count | Color | Position | Attribute | Overall (↑) |
| **Step C3: 2.5B** | 88.5 | 67.7 | 36.0 | 64.9 | 12.0 | 32.0 | 50.2 |
| **Step C4: 1.8B** | 87.5 | 62.6 | 36.5 | 68.1 | 13.0 | 31.0 | 49.8 |
| **Text Encoder Distillation** | | | | | | | |
| Model | Single Object | Two Objects | Count | Color | Position | Attribute | Overall (↑) |
| **Step C3: 2.5B** | 88.8 | 56.7 | 41.3 | 68.1 | 15.0 | 31.0 | 50.1 |
| **Step C4: 1.8B** | 87.5 | 58.7 | 40.3 | 67.8 | 12.0 | 29.0 | 49.2 |

| Teacher (12B) | 5B | 3B | 2.5B | 1.8B |
|---|---|---|---|---|

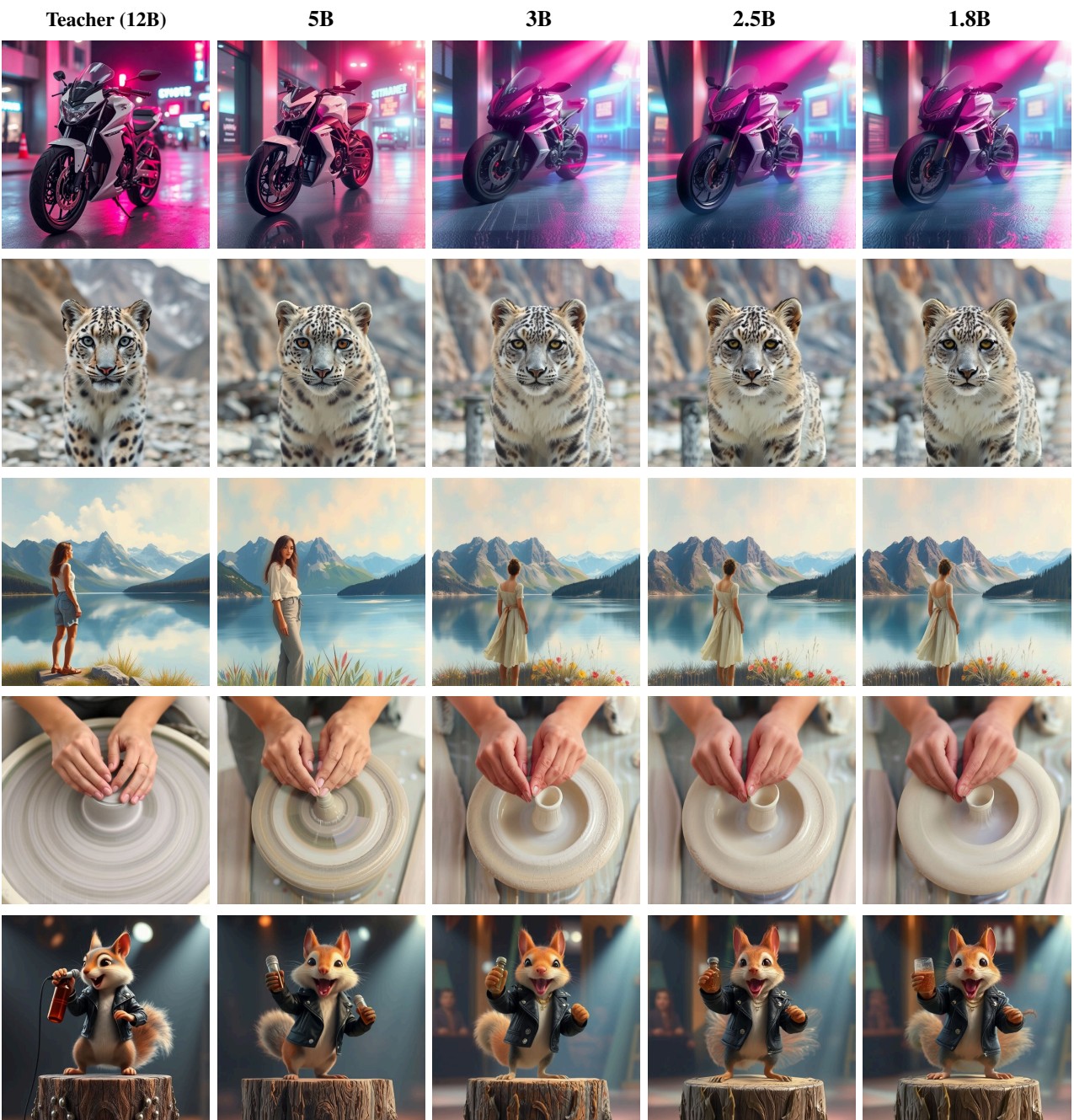

*Figure 8.* Qualitative results of the Model Compression framework (Section 4.1). We progressively distill the FLUX.1-Schnell **12B** diffusion transformer to **5B** via attention head reduction, to **3B** by reducing attention dimensionality, to **2.5B** through depth pruning, and finally to **1.8B** using Static-LN. Across all stages, the compressed models preserve the visual quality of the original model.

| FLUX.1-Schnell (17B) | SANA (4.2B) | Sana-Sprint (4.2B) | NanoFLUX (2.4B) |
|:---:|:---:|:---:|:---:|
| $(512 \times 512)$ | $(512 \times 512)$ | $(1024 \times 1024)$ | $(512 \times 512)$ |

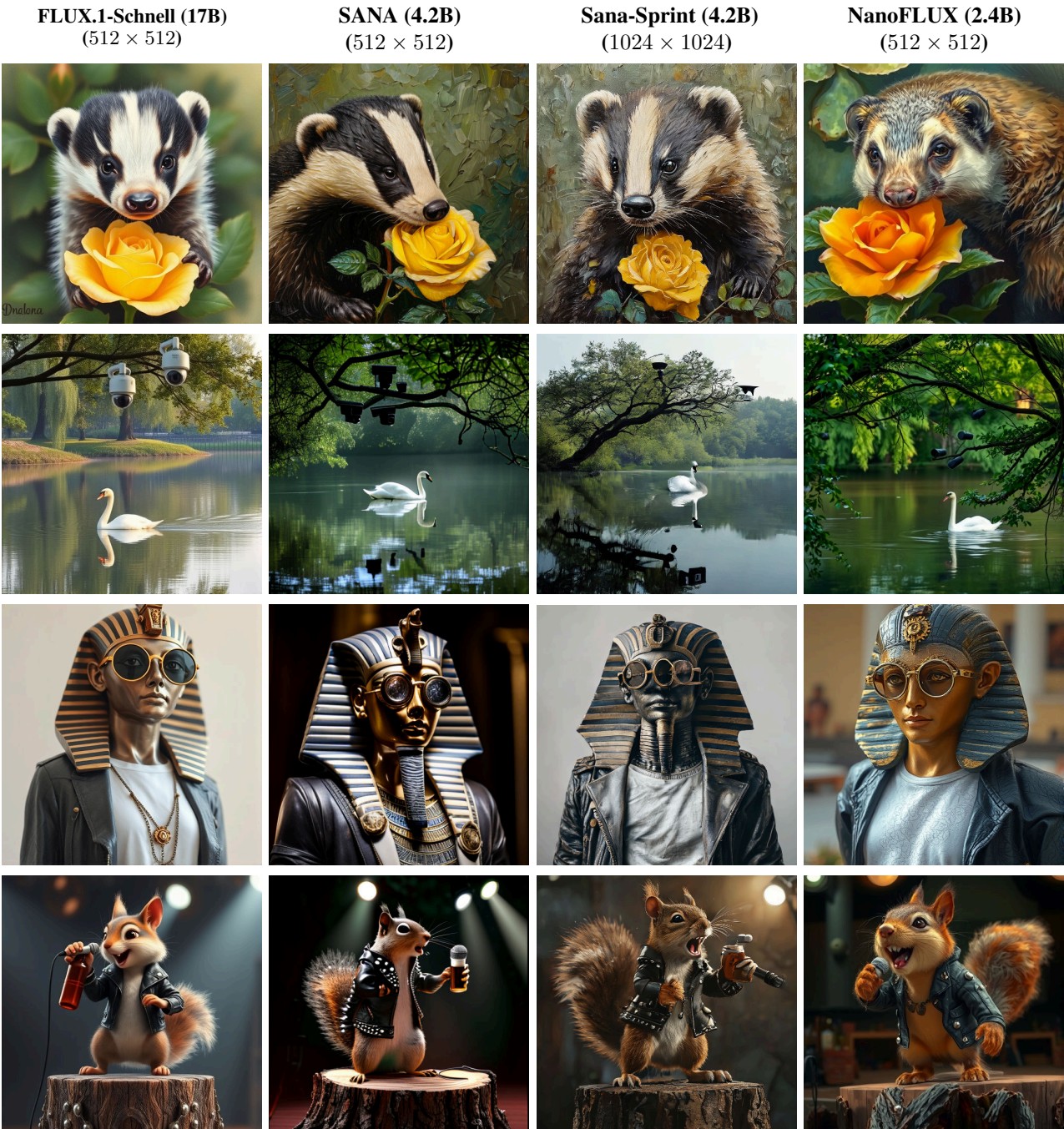

*Figure 9.* Qualitative comparison with SANA-1.5 and SANA-Sprint. NanoFLUX produces images comparable to the teacher model and SANA-1.5 and SANA-Sprint despite being half the size. Note that SANA-1.5 generates $512 \times 512$ images, while SANA-Sprint is trained only at $1024 \times 1024$ resolution and does not provide a 512 variant.

| 2.5B | 2.5B + PTD | 2.5B | 2.5B + PTD |
| --- | --- | --- | --- |

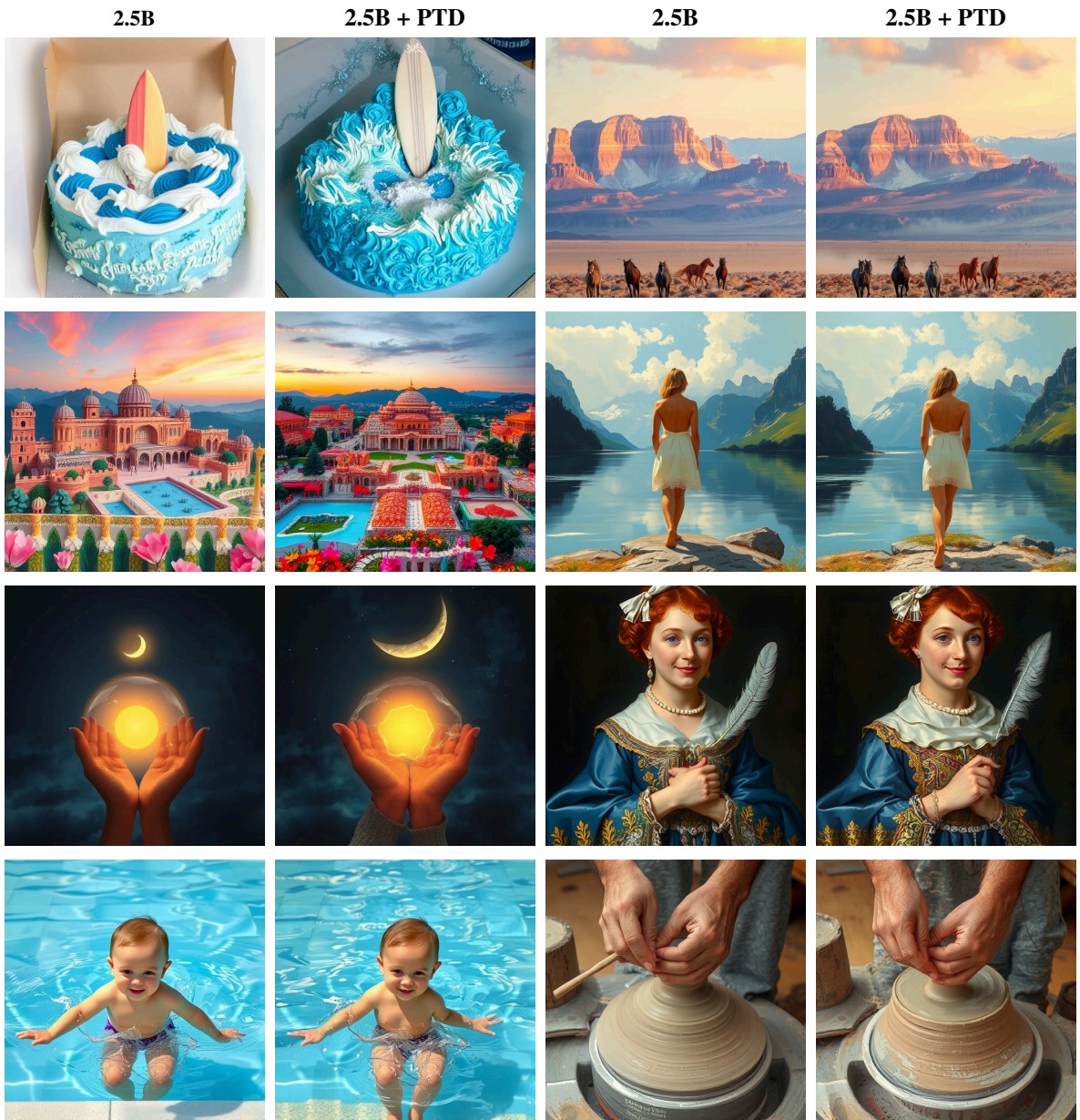

*Figure 10.* Qualitative results with Progressive Token Downsampling (Section 4.2) applied to the **2.5B** model. Token downsampling accelerates inference while preserving image quality.

