# OpenReview forum: "NanoFLUX: Distillation-Driven Compression of Large Text-to-Image Generation Models for Mobile Devices"
_ICML.cc/2026/Conference — ICML 2026 regular_

### Official Review · Reviewer_wSzp · 2026-03-11

**Soundness:** 3
**Presentation:** 2
**Significance:** 3
**Originality:** 2
**Overall Recommendation:** 4
**Confidence:** 4

**Summary:**

This paper proposes NanoFLUX, a compact text-to-image flow-matching model distilled from 17B FLUX.1-Schnell for on-device image generation. The method adopts a progressive compression pipeline to reduce the original large model into a 2.4B model while preserving generation quality. Experiments show that the final model can generate 512×512 images in about 2.5 seconds on mobile devices.

**Compliance With Llm Reviewing Policy:**

Affirmed.

**Final Justification:**

Thanks to authors for their efforts, my concerns have been addressed. I will keep my rating.

**Key Questions For Authors:**

My main concerns are about (1) the relative performance of the proposed model compared with existing compact baselines, and (2) the lack of analysis on the limitations of the compression strategy. Although the paper is meaningful from a practical perspective and the mobile deployment result is encouraging, the current empirical evidence is still not strong enough. In particular, I recommend adding comparisons with SANA 0.6B and SANA 1.6B, providing more qualitative comparisons, and discussing whether the pruning results are sensitive to the training data.

**Limitations:**

yes

**Strengths And Weaknesses:**

### Strengths

1. This paper provides a relatively systematic analysis of which components in FLUX can be compressed, and the overall compression pipeline is clear.
2. The final model can be deployed and run on mobile devices, which makes the work practically valuable.

### Weaknesses

1. Based on the reported metrics, the model seems weaker than SANA 0.6B and SANA 1.6B. It would be better to also include comparisons with these methods, as well as more analysis of the quality-efficiency trade-off.
2. The effectiveness of the compression strategy may depend heavily on the training data, but the paper does not analyze this aspect.
3. There are too few qualitative comparisons with existing methods. More visual results would help better demonstrate the strengths and limitations of the proposed model.
4. The supported resolution of 512×512 is still relatively limited.

---

> ### Author Rebuttal · Authors · 2026-03-31
>
> Thank you for your insightful review. We address your concerns below.
>
> **1. Quality-Efficiency trade-off with SANA and SANA Sprint:** Thank you for this point. We present a quality-efficiency tradeoff analysis below.
> - **On-device Latency:** SANA-Sprint employs a Deep Compression Autoencoder (DCAE) with 32× compression (F=32), compressing 1024×1024 images to 32×32 latents. In contrast, FLUX.1-Schnell and NanoFLUX utilize the FLUX-VAE, which operates at 8× spatial compression, enabling better reconstruction but larger latent sizes. For fair comparison, we adopt the VAE introduced in Diffusion-4k [1] , which achieves 16× spatial compression (VAE-F=16) in a training-free manner. This is implemented by setting the stride of the input convolution layer to 2 and applying bilinear upsampling prior to the final decoder layer. As shown in their paper (Table 2), these tweaks preserve reconstruction/generation at high resolutions for FLUX, suggesting consistency with NanoFLUX.
>
> When FLUX-VAE-F=16 is combined with patch size = 2 used in FLUX, the effective spatial compression matches that of SANA-Sprint’s DCAE. Specifically, for a 1024×1024 input, the VAE=F=16 produces 64×64 latents, which are further reduced to 32×32 (by patch size 2), resulting in the same latent resolution/compression as SANA-Sprint. The table below reports on-device latency for SANA-Sprint (1024×1024), NanoFLUX (512×512 with the original FLUX-VAE), and NanoFLUX (1024×1024 with VAE-F=16). Using VAE-F=16 at 1024×1024 produces 32×32 latent representations, which is equivalent to the latent size obtained with VAE-F=8 at 512×512. As a result, both configurations result in the same latency.
>
> |Model |SANA-Sprint|NanoFLUX|NanoFLUX|
> |-------|--------|--------|------|
> |Resolution|1024|512|1024|
> |Compression|32|F=8,p=2| F=16,p=2|
> |Steps|2|10|10|
> |Latency (ms, per-step)|  240| 240| 240|240|
> |Latency (s, total)| 0.48| 2.4| 2.4|
>
> **Quality:**
> - User study: We conducted a user study to evaluate prompt adherence, involving 5 participants who assessed a total of 150 images. These prompts were randomly picked from Geneval, DPG-Bench, and HPDv3. For each prompt, participants selected the model they preferred based on prompt adherence, with an “neither” option available if neither output adhered to the prompt. We report that NanoFLUX was preferred in 52% of cases, compared to 43% for SANA Sprint, with 5% marked as neither. These results demonstrate that NanoFLUX produces images that are preferred by human evaluators in terms of prompt adherence and aesthetics. **While SANA-Sprint attains lower latency primarily through step distillation, our user study demonstrates that this comes with a reduction in human preference.**
>
> - Failure cases of metrics: Our error analysis revealed that object detectors in GenEval fails to detect off-center objects, leading to lower scores. Furthermore, unlike the One-IG benchmark, we did not filter text-rendering prompts from DPG, which could explain the drop for DPG. Given the 7x reduction in parameter count and no training data for text rendering, we believe some degradation in text rendering is expected. However, NanoFLUX is better than SANA-Sprint for Human Preference Score (HPS) and OneIG, suggesting that the fidelity and aesthetic quality of NanoFLUX is better.
>
> In summary, we analyse the quality–efficiency trade-off between NanoFLUX and SANA-Sprint, showing that NanoFLUX achieves a more optimal balance between image quality and latency.  We will explore step distillation for NanoFLUX for future work, which could further reduce latency. However, our current focus is on large-scale architectural distillation. We will add these results in our main paper.
>
> [1] Diffusion-4K: Ultra-High-Resolution Image Synthesis with Latent Diffusion Models, Zhang et al, CVPR 2025.
>
> **2. NanoFLUX with 1024 x 1024 resolution:** Please refer to our response to Reviewer 68WZ (omitted here due to space constraints)
>
> **3 Sensitivity to training data**
> Thank you for this suggestion. We analyze the effect of the training dataset on depth pruning (Section 4.1.2). However, fully repeating the training pipeline from scratch is infeasible within the rebuttal timeframe. To study this, we consider two data distributions: our synthetic Yepop dataset (900K samples) and the open-source JourneyDB dataset (4M samples). We subsample JourneyDB to match the size of our dataset for a fair comparison.
>
> The results below show that NanoFLUX can be effectively trained using open-source data such as JourneyDB as well. Moreover, combining synthetic and real-world datasets could further improve performance, which we plan to explore in future work. At present, our results indicate that practitioners can reliably use datasets like JourneyDB to distill large-scale models using our proposed methods.
>
> | Dataset | JourneyDB|Synthetic YePop|
> |---|---|---|
> |HPS|10.94|10.68|
>
> We have addressed all your comments. We sincerely hope that you consider revising the score accordingly.

---

> > ### Author Rebuttal · Reviewer_wSzp · 2026-04-04
> >
> > Thanks to authors for their efforts, my concerns have been addressed.

---

### Official Review · Reviewer_68WZ · 2026-03-13

**Soundness:** 3
**Presentation:** 3
**Significance:** 3
**Originality:** 3
**Overall Recommendation:** 4
**Confidence:** 5

**Summary:**

This paper proposes NanoFLUX, a distillation and compression method that turns the 17B FLUX.1-Schnell T2I model into a 2.4B model that can run on a mobile device. First, the progressive pipeline compresses the 12B MM-DiT through head pruning, head-dimension reduction, block merging and a static normalization variant. Second, progressive token downsampling is explored to improve throughput efficiency. Finally, the 5B T5-XXL text encoder is distilled into a 330M T5-Large model. NanoFLUX can generate 512×512 images in about 2.5 seconds on a Samsung S25 Ultra.

**Compliance With Llm Reviewing Policy:**

Affirmed.

**Final Justification:**

The rebuttal addressed my concerns. Therefore, I increase the score to acceptance

**Key Questions For Authors:**

- Could you provide more baseline for model with comparable params? Table 4 does not include a genuinely parameter-comparable external baseline in the ~2–3B total range.
- Missing baseline for mobile (e.g. MobileDiffusion 386M, SnapFusion 848M).
- Could the authors report latency measured for baselines on the same device for the models in Table 4? The paper provides on-device latency for internal ablations, but without same-device measurements for external baselines.
- Wang et al. (2025)’s supplementary already evaluates replacing PixArt-Alpha’s T5-XXL with T5-Base. Since T5-Base has about 220M parameters, that setup is roughly 0.82B total parameters by simple addition. Could the authors discuss how NanoFLUX compares against such a smaller text-encoder replacement baseline, especially in terms of efficiency-quality tradeoff?
- Wang et al. (2025) is a closely related method that also studies replacing T5-XXL with smaller T5 encoders. However, the comparison in Table 10 appears somewhat limited, and the results seem marginal. Could the authors provide a more comprehensive comparison, including metrics such as DPG and GenEval, where the gap to the original model seems more noticeable?
- What image resolution is used for the models in Table 4? To my understanding, SANA-Sprint is only available in 1024×1024 variants. If Table 4 reports results from the 1024×1024 version of SANA-Sprint, then the comparison may not be apples-to-apples with 512×512 NanoFLUX
- Can the distilled T5 proposed in this work be plugged into other model families, such as SD3 or PixArt-Alpha? Demonstrating such transferability would help clarify whether the distilled encoder is broadly useful or mainly tailored to the FLUX family ?

**Limitations:**

yes

**Strengths And Weaknesses:**

### Strengths:
- The paper addresses a relevant problem: how to bring strong T2I models closer to real on-device use. This is an important direction, and the paper has clear practical motivation.

- Clear staged methodology with interpretable components and appendix support.

- The ablations are useful. The comparisons for block merging, PTD, and text encoder distillation help show that the system was not built completely heuristically.

### Weakness:

- The final claim about preserving quality feels strong. In Table 4, NanoFLUX is still meaningfully below FLUX.1-Schnell on DPG and GenEval, with 2.5x increasing of NFEs.

- The text encoder distillation story is interesting, but the evaluation is still incomplete. The paper shows that distilling T5-XXL to T5-Large works inside this FLUX family, but it is unclear whether the distilled encoder would also be useful in another T5-based text-to-image model.

- This is a good engineering paper, but the empirical evaluation is still missing several important baselines, such as models with comparable parameter counts in Table 4 and additional mobile text-to-image baselines.

---

> ### Author Rebuttal · Authors · 2026-03-31
>
> Thank you for your review. We address your concerns below.
>
> **1. Quality preservation, DPG/Geneval metrics:** We argue that "quality preservation" for on-device models like NanoFLUX should be viewed through the lens of quality-efficiency Pareto frontier instead of quality in isolation. In this regard, NanoFLUX offers a significantly better trade-off for on-device image generation. Please have a look at trade-off analysis for SANA Sprint. Regarding Geneval/DPG and NFEs:
> - While we increase the NFEs by 2.5x, with 7x architectural compression, the overall latency of NanoFLUX is still significantly below the teacher model (Table 6). We will explore step distillation into 4 steps in future work.
> - Our error analysis revealed that object detectors in GenEval occasionally fail to detect off-center objects, leading to lower scores that do not reflect a loss in alignment. Furthermore, unlike the One-IG benchmark, we did not filter text-rendering prompts from DPG, which explains the drop for DPG. At the same time, given the 7x reduction in parameter count and no training data for text-rendering, some degradation in text rendering is still expected. However, the fact that NanoFLUX is comparable for Human Preference Score (HPS) and OneIG suggests that the fidelity and aesthetics are preserved.
>
> **2. Transferrability of text encoder:** In our paper, we distilled the T5-Large model with intermediate textual embeddings from our 2.5B model (section 4.1.2). Following a setup similar to Wang et al, we validate the transferability of T5-Large trained with FLUX.1-Schnell (12B) by evaluating it with the FLUX.1-Dev (12B) model.
> | Text-encoder | One-IG |  DPG | Geneval | HPS|
> |--| --|--|----|---|
> |T5-Large| 52.1 | 82.7 | 64.7 | 11.78|
> |T5XXL |53.5 | 83.8 |  67.0 | 12.01|
>
> **3. Missing baselines (MobileDiffusion 386M, SnapFusion 848M):** At the time of submission, code/pretrained weights for these models were not available, preventing local evaluation. While both report FID on MS-COCO30k, we do not include this metric as the dataset distribution differs significantly from the high-res aesthetic images of FLUX.1-Schnell and NanoFLUX. Instead, we compare on-device latency and quality–efficiency trade-offs against SANA-Sprint.
>
> **4. Comparison with SANA-Sprint:** Please refer to our response to Reviewer wSzp. We present on-device latency for SANA-Sprint. However, we observe a latency-quality trade-off, where latency comes at a cost of human preference in quality for SANA Sprint.
>
> **5. Small text-encoder replacement:** We repeat our text encoder distillation experiments with T5-Base and 2.5B model as the diffusion model and compare with Wang et al.
> |Method|One-IG|DPG|Geneval|HPS
> |--|--|--|--|--|
> |Ours | 41.3 | 73.7 | 40.9 | 10.26 |
> |Wang et al | 40.5 | 72.9 | 40.8 | 10.12 |
>
> **6. Missing DPG-Bench and Geneval for Table 10:** Results with 2.5B model for DPG and Geneval for text-encoder ablations in Table 10 are presented below. Our method performs better than Wang et al's.
>
> | Method | DPG | Geneval |
> |---|--|--|
> |Projection loss| 70.3 | 42.85|
> |Wang et al | 72.1 | 46.47|
> |Ours| 76.3 | 50.12|
>
> **7. NanoFLUX with 1024 x 1024 resolution:** We trained NanoFLUX at 1024 x 1024 resolution using FLUX.1-Schnell as the teacher. Every iteration was trained with a lr=1e-4 for 20 epochs and a batch size of 16 on 64 H00 GPUs. As shown in the table below, NanoFLUX achieves performance comparable to FLUX.1-Schnell. Furthermore, when compared to the similarly sized SANA-Sprint model, NanoFLUX attains a lower (better) average rank of 1.25 vs 1.75 for SANA-Sprint.
>
> |Model|One-IG | DPG | Geneval | HPS|
> |-------|-------|-------|-------|-------|
> |SANA Sprint| 43.2 |63.9 | 73.0 | 10.26|
> |FLUX.1-Schnell |48.3 | 83.2 | 66.5 | 12.14|
> |NanoFLUX (Ours)|44.8 | 77.6 | 52.7 | 10.49|
>
> **8. Can our distillation method be plugged into other model families, such as SD3 or PixArt-Alpha?** Following Wang et al., we distill T5-Large/Base using representations derived from a diffusion model. However, we note that Wang et al also focuses on model-specific text encoder distillation, rather than transferability across different model families. Nevertheless, to address the reviewer’s question, we show in the table below our distillation approach generalizes to other diffusion models, such as SDv3.5, when used during training.
>
> We distill a T5-Base model from the SDv3.5 T5-Large encoder. The student model is first trained during a warm-up stage for 2 epochs with a learning rate of 1e-4, followed by 20 epochs using our prompt-embedding-based loss at a learning rate of 1e-5. We also train T5-Base using the method (and hyper-params) of Wang et al. for 50k iterations (batch size = 32) with a learning rate of 1e-4.
>
> | Method| One-IG |  DPG | Geneval | HPS|
> |-------| -----|-----|------|------|
> |Wang et al| 41.47 | 73.68 | 41.76|9.37|
> |Ours |42.65 | 76.17 | 43.27 | 9.48|
>
> We have addressed all comments and sincerely hope that you consider revising the score accordingly.

---

> > ### Author Rebuttal · Reviewer_68WZ · 2026-04-03
> >
> > Thanks for resolving my concern. I will increase the score.

---

### Official Review · Reviewer_hrVg · 2026-03-16

**Soundness:** 3
**Presentation:** 4
**Significance:** 4
**Originality:** 3
**Overall Recommendation:** 4
**Confidence:** 3

**Summary:**

This paper studies model parameter and compute compression in diffusion transformers, especially, the parameters of the diffusion transformer network per se and the text encoder, from the distillation view. This paper starts from the step-by-step analysis of parameter counts, and conduct extensive progressive parameter pruning with fruitful intermedia results. This paper also considers a token downsampling technique such that tokens in most diffusion transformer layers are reduced as the computing need. As a result, the distilled variant (1.8B or 2.5B) can run on mobile devices and achieve competitive results.

**Compliance With Llm Reviewing Policy:**

Affirmed.

**Key Questions For Authors:**

1. On line 204, when mentioned training images recaptioning, the authors specified Qwen-Image (Wu et al., 2025a) to generate various caption given an image. But as my best knowledge, Qwen-Image is mainly for visual generation rather than for visual captioning. Could authors please double check this point to include more information or see whether the citation is wrong? (also on line 609)

**Limitations:**

Yes.

**Strengths And Weaknesses:**

Strengths:

1. This is overall a comprehensive and solid paper on where we can reduce parameter counts and computational costs in large diffusion models. This paper studies extensively attention head pruning, attention dim pruning, redundant transformer layer pruning, token reduction, text encoder pruning, and transforming AdaLN into static ones. The author shows the model performance after every modification step and the experiment designs are sound and insightful. The finally-resulted NanoFLUX model can run on Samsung S25U and reach the competitive result compared to the teacher model. The detailed experiment results and their detailed analysis should be encourage as a conference paper, which adds many positive points to this paper. In other words, this paper can serves as a manual for distillation researchers and engineers in the future.

Weaknesses:

1. This paper lacks a detailed analysis of VRAM/RAM details for inference on mobile devices, including loading model per se and maximum activation VRAM use. Since this distilled model is targeted for mobile use, the inference VRAM/RAM details should be noted.

2. The most vulnerable part of this paper, in my opinion, is the weak originality of proposed methods in this paper. Though being detailedly experimented and reported, the technical part of this paper seemed to be too engineering with a practical goal while scientific new methods/perspective are absent, since every component of this paper has been already studied in a paper. In other words, this paper only describe combinations of existing methods. But after all, such detailed experiment results and analysis should be encouraged and a merit.

---

> ### Author Rebuttal · Authors · 2026-03-30
>
> Thank you for your positive review, and acknowledging the comprehensiveness of our work that could help distillation researchers and engineers in the future. We address your concerns below.
>
> **1. On-device Model loading and maximum activation VRAM:** Loading the model on the Samsung S25 Ultra takes approximately 2 seconds. This can further be optimized in future works. The maximum activation VRAM is 2GB as calculated by Snapdragon profiler. Thank you for pointing this out, we will add this in the main paper.
>
> **1. Engineering vs Academic research:** We thank the reviewer for recognizing the depth of our analysis. However, we respectfully disagree with the characterization that our work is merely a combination of existing methods. While certain components (e.g., attention layer pruning in Section 4.1.1 and depth pruning in Section 4.2.2) have prior precedent, their combination is not a straightforward integration. Additionally, our contributions go beyond the intelligent way of leveraging existing pruning strategies and we have introduced the following novel techniques with both practical and scientific goals.
> - Static LayerNorm: First training-free approach that achieves up to 30% model size reduction by precomputing adaptive normalization coefficients.
> -  ResNet-based Progressive Token Downsampling: We introduce a new architectural design and training strategy using hybrid resolutions, demonstrating that progressive token reduction can substantially reduce latency while preserving generation.
> - Prompt-Embedding-Based Text Encoder Distillation: We propose a novel distillation framework that leverages prompt embeddings from intermediate transformer blocks to transfer visual-semantic knowledge from large text encoders to compact models.
>
> Finally, we emphasize that NanoFLUX presents a large-scale distillation framework where each step is motivated by systematic analysis and findings that we consider to be both engineering and scientific contributions. So while our work is motivated by practical efficiency, its primary contribution is not the engineering itself, but the insights and techniques that practioners can use from this paper.
>
> **3. Regarding Qwen-Image citation** - Thank you for pointing this out. We used Qwen2-VL-7B-Instruct for dataset generation, not Qwen-Image. We will correct this mistake in the paper.
>
> We have addressed all your comments. We sincerely hope that you consider revising the score accordingly.

---

> > ### Author Rebuttal · Reviewer_hrVg · 2026-04-04
> >
> > Thanks for the author rebuttal. I will maintain my positive score therefore.

---

### Decision · Program_Chairs · 2026-04-30

**Decision:**

Accept (regular)

**Comment:**

The paper presents NanoFLUX, a compressed text to image generation system distilled from a larger `FLUX.1-Schnell` model into a 2.4B model for mobile deployment. The method uses a progressive compression pipeline that reduces the larger diffusion transformer through attention pruning, head dimension reduction, block merging, and replacement of adaptive normalization with static normalization. It further introduces progressive token downsampling to reduce latency and distills the larger T5 text encoder into a smaller T5-Large model. The submission reports that the final system can generate 512 x 512 images in about 2.5 seconds on mobile devices, and it supports the approach with staged ablations across the major compression steps and evaluations on One IG, DPG, GenEval, and HPSv3.

The strengths of the submission are its strong practical motivation, clear decomposition of the overall system into understandable stages, and the unusually thorough empirical study of where compression can be applied in a large text to image model. The reviewers note that the work is engineering focused and that some comparisons and analyses could be expanded, including memory usage on device, broader baseline coverage, and more discussion of quality tradeoffs. I agree that the novelty is more incremental than conceptual, and that the final model does not fully match the teacher on all metrics. However, I believe the paper still makes a meaningful contribution because it turns a large model family into a realistic on device system, provides a careful and reproducible study of multiple compression decisions, and offers insights that are likely to be useful to both researchers and practitioners. Additional empirical datapoints provided in the rebuttal against SANA also helps to improve the usefulness of the proposed approach and the trained model. The combination of practical significance, systematic ablations, and strong experimental results gives the paper enough merit for acceptance in my opinion.